# ONE-SHOT EXEMPLARS FOR CLASS GROUNDING IN SELF-SUPERVISED LEARNING

**Haowen Cui**[1]   **Shuo Chen**[2*]   **Jun Li**[1,2]   **Jian Yang**[1,2*]
[1] PCA Lab, Nanjing University of Science and Technology
[2] School of Intelligence Science and Technology, Nanjing University

## ABSTRACT

Self-Supervised Learning (SSL) has recently achieved remarkable progress by leveraging large-scale unlabeled data. However, SSL pretrains models without relying on human annotation, so it usually does not specify the class space. This inevitably weakens the effectiveness of the learned representation in most downstream tasks that have the intrinsic class structure. In this work, we introduce the new easy setting of One-Shot Exemplar Self-Supervised Learning (OSESSL), requiring only one instance annotation for each class. By introducing this extremely sparse supervision, OSESSL provides the minimum class information to guide the exploration of unlabeled data, achieving significant performance boosts with neglectable annotation cost (i.e., a complexity of $\mathcal{O}(1)$ w.r.t. the sample size). In this OSESSL setting, we propose a simple yet effective framework that leverages the single-labeled exemplar to build the class-specific prototype for learning reliable representations from the huge unlabeled data. To this end, we also build a novel consistency regularization, which extends the sparse exemplar supervision into the decision boundaries, thus improving the robustness of the learned representation. Extensive experiments on real-world datasets clearly validate the reliability of this simple and practical setting. The proposed approach successfully outperforms the state-of-the-art methods, achieving gains of approximately 3% and 6% $k$-NN accuracy on CIFAR-100 and ImageNet-100, respectively.

## 1 INTRODUCTION

Learning generalizable visual representations has been a long-standing challenge. In recent years, Self-Supervised Learning (SSL) (Chen et al., 2020a; He et al., 2020b; Grill et al., 2020; Caron et al., 2020; 2021; Oquab et al., 2024) has emerged as a powerful unsupervised paradigm, demonstrating impressive success in various downstream tasks such as classification, detection, and segmentation (Azizi et al., 2021; Wang et al., 2021; Tang et al., 2022; Zou et al., 2022). By leveraging large-scale unlabeled datasets, SSL aims to discover semantic structures through auxiliary objectives that replace explicit labels.

Clustering-based approaches have emerged as the influential methods in SSL. Representative methods include SwAV (Caron et al., 2020) and DINO (Caron et al., 2021), which achieve strong performance by aligning the cluster predictions of different augmented views of the same image. However, these methods do not specify the class space for the model to learn towards. It is inevitable that the effectiveness of the learned representations is limited in downstream tasks with the corresponding intrinsic class structure, and the emergent clusters are hardly guaranteed to align with the true classes. This also echoes the fundamental insight from machine learning known as the *no-free-lunch* theorem (Wolpert & Macready, 2002).

To address this limitation, we introduce the One-Shot Exemplar Self-Supervised Learning (OSESSL), which assumes access to only a single annotated instance per class. These annotations serve as exemplars that expose the true class space and provide the minimal yet crucial supervision. As shown in Fig. 1, this guidance enables the learned representations to be steered toward meaningful semantic structures while retaining the scalability of SSL. Meanwhile, recent studies (Chen et al., 2021b;

---

*Corresponding Authors

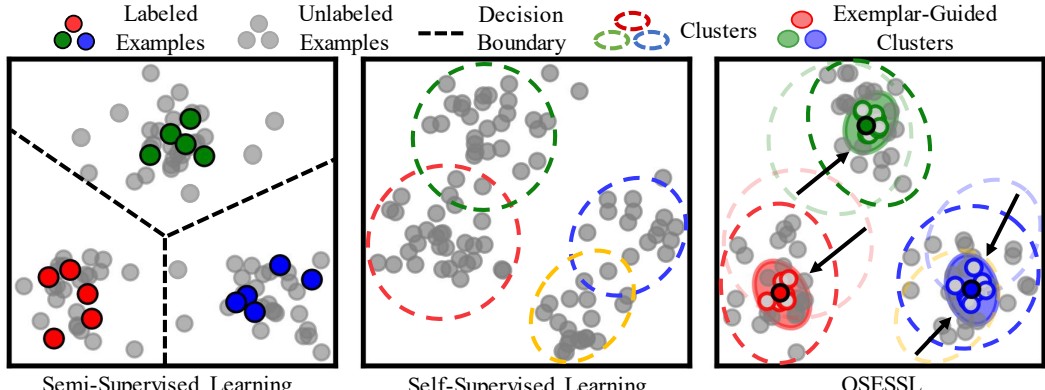

Figure 1: Illustration of different learning paradigms. OSESSL can lead to semantically compact representations with only one annotation per class.

Lu et al., 2022; Dong et al., 2023) have demonstrated the effectiveness of incorporating SSL into few-shot learning, which further motivates our exploration of SSL in the extreme one-shot regime. Importantly, since the number of classes typically grows much slower than the sample size in practical scenarios (Schuhmann et al., 2022; Dehghani et al., 2023), the required annotations are actually negligible compared to the data scale, keeping the annotation complexity effectively at $\mathcal{O}(1)$ with respect to the sample size. Under this setting, the challenge lies in effectively propagating supervision from such extremely sparse data to the unlabeled data without overfitting or collapsing.

To solve this new problem setting, we propose a simple yet effective method that incorporates extremely sparse supervision into SSL. Specifically, we first construct exemplar-guided prototypes by augmenting each exemplar with its most representative neighbors from the unlabeled data. This ensures that the prototypes are both semantically grounded in the actual classes and sufficiently representative of the data distribution, establishing a reliable basis for representation learning. We then leverage these prototypes to align augmented views of unlabeled data, facilitating knowledge transfer from the annotated exemplars to the unlabeled majority. To prevent prototype collapse, we explicitly enforce a prototype dispersion regularization, which promotes diversity among the constructed prototypes. Additionally, we incorporate an exemplar-guided interpolation consistency that regularizes the model on mixed embeddings. This smooths decision boundaries in regions where exemplar guidance is less certain, thereby enabling the more reliable knowledge transfer and stronger generalization. Extensive experiments on standard SSL benchmarks demonstrate that our approach achieves state-of-the-art performance, showing strong generalizability and practical effectiveness.

The main contributions of this work are summarized as follows:

- We explore a new SSL setting that exploits a single annotation per class to provide semantic exemplars, offering a very practical balance between annotation cost and model performance.
- We propose a novel framework combining exemplar-guided prototype learning and exemplar-guided interpolation consistency to propagate the infrequent supervisory signal and ensure the robustness of the model.
- Comprehensive experimental results on CIFAR and ImageNet demonstrate the superiority of our approach over the state-of-the-art methods.

## 2 RELATED WORKS

### 2.1 SELF-SUPERVISED LEARNING

SSL has emerged as a powerful paradigm for representation learning. The main challenge of SSL is representation collapse, where all samples are encoded to a constant output. To mitigate this problem, a variety of paradigms have been developed.

Recent advances have been largely driven by contrastive learning, such as SimCLR, MoCo, BYOL and SimSiam (Chen et al., 2020a; He et al., 2020b; Chen et al., 2020c; Grill et al., 2020; Chen &

He, 2021). They enforce consistency between embeddings with different augmentations of the same instance while preventing collapse through the usage of negatives, architectural asymmetry or the stop-gradient mechanism. Furthermore, recent advancements in feature decorrelation (Zbontar et al., 2021; Ermolov et al., 2021b; Bardes et al., 2022; Weng et al., 2024) introduce principled regularization terms that explicitly minimize redundancy among feature representations. Another direction of research focuses on clustering-based methods, such as SwAV (Caron et al., 2020), DINO (Caron et al., 2021), ReSA (Weng et al., 2025) and SOP (Silva et al., 2025). They simultaneously learn feature representations and cluster assignments, aligning embeddings with prototypical representations. Such methods implicitly combine contrastive and predictive principles by encouraging consistency within clusters while preserving discriminative power.

While SSL has achieved remarkable success, its representations are learned from self-generated supervisory signals that may not align with true semantic classes. This gap between the feature space and human-defined categories motivates our work. We propose to incorporate minimal labeled data to guide the learning process, grounding representations in real semantic categories while preserving the scalability and robustness of SSL.

## 2.2 Self-Supervised Learning with Annotations

Several works have explored incorporating supervision into self-supervised frameworks. SupCon (Khosla et al., 2020) and ANCL (Oh & Lee, 2024) extend contrastive learning by exploiting label information to pull together embeddings of the same class, improving the corresponding performance when sufficient annotations are available. Recent semi-supervised learning methods have been strengthened by integrating self-supervised objectives, thereby exploiting the representation ability of SSL to boost the performance of semi-supervised learning. SimCLRv2 (Chen et al., 2020b) verifies that the task-agnostic self-supervised learning is effective for semi-supervised learning. PAWS (Assran et al., 2021) extends the self-supervised loss to the semi-supervised learning via non-parametrically predicting view assignments with labeled data. SsCL (Zhang et al., 2022) integrates the contrastive loss into semi-supervised learning with similarity co-calibration. Suave (Fini et al., 2023) unifies a supervised objective using ground-truth labels and a self-supervised objective based on clustering assignments into a single optimization process via a cross-entropy loss.

These methods usually assume access to a sufficient portion of labeled data, which is proportional to the sample size. In contrast, our method requires only a single annotated sample per class, making the annotation cost essentially independent of the sample size. This arises because in practical scenarios the class size much more slowly than the sample size, so the required annotations remain negligible even as the dataset scales (Schuhmann et al., 2022; Dehghani et al., 2023). By constructing exemplar-guided prototypes from sparse annotations and propagating their supervision to unlabeled samples, our framework achieves semantically consistent and robust representation learning with minimal annotation effort.

It is important to distinguish OSESSL from conventional semi-supervised or few-shot learning. Semi-supervised learning uses labels to directly train classifier boundaries. Few-shot methods focus on learning to classify unseen categories. In contrast, OSESSL uses a single exemplar per class as semantic anchors to expose the class space and guide representation learning under predominantly unlabeled samples. This minimal class grounding directly addresses the mismatch between self supervision and semantic class structures. This minimal supervision is required to overcome the "no-free-lunch" limitation of self-supervised learning.

## 3 Methods

### 3.1 Preliminary

Given an unlabeled dataset $\mathcal{D}_u = \{\mathbf{x}_u^{(1)}, \mathbf{x}_u^{(2)}, ..., \mathbf{x}_u^{(N)}\}$, we denote by $\mathbf{b}$ a mini-batch of $n$ unlabeled images sampled from $\mathcal{D}_u$. For each input mini-batch, we use a pair of random transformations $\mathcal{T}$ and $\mathcal{T}'$ to generate two augmented views $\mathbf{x} = \mathcal{T}(\mathbf{b}) \in \mathbb{R}^{n \times 3 \times H \times W}$ and $\mathbf{x}' = \mathcal{T}'(\mathbf{b}) \in \mathbb{R}^{n \times 3 \times H \times W}$, where $H$ and $W$ are the height and width of the image. Let $\mathbf{f}$ and $\mathbf{f}'$ denote a pair of neural networks, each consisting of a feature encoder ($\mathbf{e}$ or $\mathbf{e}'$) and a projector ($\mathbf{g}$ or $\mathbf{g}'$). They encode the augmented views to normalized $d$-dimensional embedding features, i.e., $\mathbf{z} = \mathbf{g}(\mathbf{e}(\mathbf{x})) \in \mathbb{R}^{n \times d}$ and $\mathbf{z}' = \mathbf{g}'(\mathbf{e}'(\mathbf{x}')) \in \mathbb{R}^{n \times d}$.

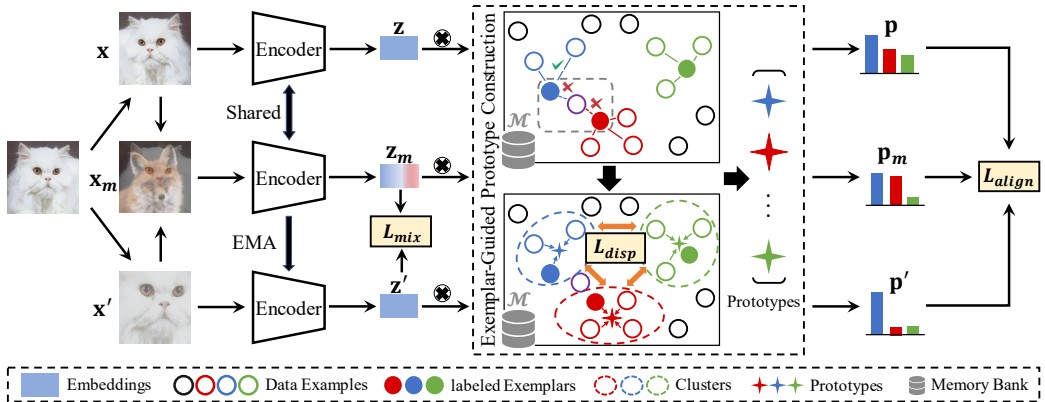

Figure 2: Overview of our proposed method. Given augmented views and mixed view, each view is encoded by the encoder resulting in embeddings. The labeled exemplars are leveraged to construct prototypes for each class with their discriminative neighbors. The alignment objective $\mathcal{L}_{\text{align}}$ enforces the distributions of different views to be consistent, while $\mathcal{L}_{\text{disp}}$ encourages the prototypes to remain diverse. $\mathcal{L}_{\text{mix}}$ ensures the consistency between mixed view and original views.

For clustering-based self-supervised learning, two logits are computed by measuring the similarity between embeddings and a set of prototypes. The prototypes can be either parameterized via a learnable projection layer (Caron et al., 2020; 2021), or dynamically constructed from inter-sample relations (Weng et al., 2025). To transform logits into probability distributions (cluster assignments), one typically applies a mapping function, which can be instantiated as either a softmax function or the Sinkhorn-Knopp algorithm (Cuturi, 2013). These assignments are then used to encourage consistent predictions across augmented views, formulated by the following cross-entropy loss:

$$\mathcal{L}_{\text{cluster}} = -\frac{1}{n} \sum_{i=1}^{n} \mathbf{t}^{(i)} \log(\mathbf{s}^{(i)}), \tag{1}$$

where $\mathbf{s}^{(i)}$ and $\mathbf{t}^{(i)}$ are the assignment distributions of the $i$-th sample.

## 3.2 ONE-SHOT EXEMPLAR SSL

In OSESSL, an unlabeled dataset $\mathcal{D}_u = \{\mathbf{x}_u^{(1)}, \mathbf{x}_u^{(2)}, ..., \mathbf{x}_u^{(N)}\}$ and a labeled dataset in the form of a single annotated image per class $\mathcal{D}_l = \{\mathbf{x}_l^{(1)}, \mathbf{x}_l^{(2)}, ..., \mathbf{x}_l^{(C)}\}$ are given, where $C$ is the number of classes of the dataset. The labeled set size scales with the number of classes in the dataset.

Our goal is to learn representations by leveraging both $\mathcal{D}_u$ and $\mathcal{D}_l$ during pretraining. Each labeled image acts as a class exemplar to the underlying class space. Here, a crucial challenge arises: how to effectively leverage the extremely sparse supervision signal to unlabeled data. To address this challenge, we introduce an exemplar-guided prototype learning mechanism that constructs prototypes from exemplars and unlabeled samples to provide reliable guidance for representation learning.

**Exemplar-Guided Prototype Construction.** To ensure the constructed prototypes are representative and generalizable beyond the extremely sparse supervision, we propose to enrich them by aggregating discriminative information from the unlabeled data. This is achieved by augmenting each exemplar with a set of carefully selected neighbors from the unlabeled pool, which in turn forms a more comprehensive and robust prototype for each class. Unlike conventional clustering-based SSL (Caron et al., 2020; 2021; Weng et al., 2025; Silva et al., 2025), our prototypes are semantically aligned with the underlying class structure, as they are explicitly grounded in the provided exemplars rather than emerging solely from unsupervised clustering.

Let $\mathbf{x}_l \in \mathbb{R}^{C \times 3 \times H \times W}$ denote the view of labeled images transformed by $\mathcal{T}$. Similar to unlabeled images, the network encodes the labeled view to the normalized embedding feature $\mathbf{z}_l = \mathbf{g}(\mathbf{e}(\mathbf{x}_l)) \in \mathbb{R}^{C \times d}$. Meanwhile, we maintain a first-in-first-out memory bank $\mathcal{M} = \{\mathbf{m}^{(1)}, \mathbf{m}^{(2)}, ..., \mathbf{m}^{(M)}\}$ that stores $M$ historical embeddings of unlabeled images. Given labeled exemplars $\mathbf{z}_l$ and the memory

bank $\mathcal{M}$, we select $k$ nearest neighbors for each class based on a discriminative score

$$s^{(c)}(j) = \alpha \langle \mathbf{z}_l^{(c)} \cdot \mathbf{m}^{(j)} \rangle - (1-\alpha) \max_{c' \neq c} \langle \mathbf{z}_l^{(c')} \cdot \mathbf{m}^{(j)} \rangle, \quad j \in \{1, \ldots, M\}, \tag{2}$$

where $\alpha \in [0, 1]$ is a weighting factor to balance the contributions of different similarity terms. This selection strategy prioritizes samples that exhibit both high similarity to intra-class exemplars and clear differentiability from inter-class exemplars. Accordingly, we define $\mathcal{S}_c$ as the set of indices for the top-$k$ neighbors in $\mathcal{M}$ ranked by $s^{(c)}(j)$. The neighbor-augmented prototype for class $c$ is them formed by combining the exemplar with the corresponding selected samples:

$$\mathcal{P}^{(c)} = \{\mathbf{q}^{(c,0)}, \mathbf{q}^{(c,1)}, \ldots, \mathbf{q}^{(c,k)}\} = \{\mathbf{z}_l^{(c)}\} \cup \{\mathbf{m}^{(j)} \mid j \in \mathcal{S}_c\}, \quad |\mathcal{P}^{(c)}| = k+1. \tag{3}$$

To further reduce the noise of false positive, we compute soft weights for each class $c$ over its $k+1$ components by measuring their similarity to the exemplar:

$$\pi^{(c,j)} = \frac{\exp\left(\langle \mathbf{z}_l^{(c)} \cdot \mathbf{q}^{(c,j)} \rangle\right)}{\sum_{j'=0}^{k} \exp\left(\langle \mathbf{z}_l^{(c)} \cdot \mathbf{q}^{(c,j')} \rangle\right)}, \quad \mathbf{q}^{(c,j)} \in \mathcal{P}^{(c)}. \tag{4}$$

Consequently, the exemplar-guided prototype for class $c$ can be computed as $\mathbf{c}^{(c)} = \sum_j \pi^{(c,j)} \mathbf{q}^{(c,j)} \in \mathbb{R}^{1 \times d}$.

**Exemplar-Guided Prototype Learning.** With the class grounded prototypes established, we want to ensure that they can guide representation learning of unlabeled data toward semantically meaningful directions. To this end, we enforce consistency between different augmented views in the prototype space via an alignment objective. To be specific, for a pair of unlabeled embeddings $\mathbf{z}$ and $\mathbf{z}'$, we calculate the probability for $i$-th sample to the $c$-th prototype

$$p^{(i,c)} = \frac{\exp\left(\langle \mathbf{z}^{(i)} \cdot \mathbf{c}^{(c)} \rangle / \tau_s\right)}{\sum_{c'=1}^{C} \exp\left(\langle \mathbf{z}^{(i)} \cdot \mathbf{c}^{(c')} \rangle / \tau_s\right)}, \quad p'^{(i,c)} = \frac{\exp\left(\langle \mathbf{z}'^{(i)} \cdot \mathbf{c}^{(c)} \rangle / \tau_t\right)}{\sum_{c'=1}^{C} \exp\left(\langle \mathbf{z}'^{(i)} \cdot \mathbf{c}^{(c')} \rangle / \tau_t\right)}, \tag{5}$$

where $\tau_s$ and $\tau_t$ are positive temperature parameters to sharpen the probability. We then align the distributions of different augmented views using the cross-entropy loss

$$\mathcal{L}_{\text{align}} = -\frac{1}{n} \sum_{i=1}^{n} \sum_{c=1}^{C} p'^{(i,c)} \log p^{(i,c)}. \tag{6}$$

This ensures that unlabeled data are not only encouraged to produce stable predictions, but also implicitly guided toward the exemplar-guided prototypes. To better understand its effect, we calculate the gradient of $\mathcal{L}_{\text{align}}$ with respect to the embedding $\mathbf{z}$. Since $p^{(i,c)}$ is a softmax over prototype similarities, the derivative of its log-probability with respect to $\mathbf{z}$ can be formulated as

$$\frac{\partial \log p^{(i,c)}}{\partial \mathbf{z}} = \frac{1}{\tau_s} \left(\mathbf{c}^{(c)} - \sum_{c'} p^{(i,c')} \mathbf{c}^{(c')}\right). \tag{7}$$

Combining Eq. (6) and Eq. (7), we can further obtain that

$$\nabla_{\mathbf{z}} \mathcal{L}_{\text{align}} = \left(\mathbb{E}_p[\mathbf{c}] - \mathbb{E}_{p'}[\mathbf{c}]\right)/\tau_s, \tag{8}$$

where we can observe that each unlabeled embedding is pulled toward the prototype barycenter prescribed by the exemplar-guided target distribution. Consequently, the alignment indeed propagates sparse exemplar supervision to unlabeled data, steering representation learning toward the correct semantic direction. This differs from clustering-based methods, which lacks explicit class grounding.

Minimizing the above loss can provide explicitly direction for representation learning. However, the alignment can not prevent prototype collapse, where different classes may converge to the similar representation. To avoid this, we add a contrastive-style dispersive regularization:

$$\mathcal{L}_{\text{disp}} = \frac{1}{C(C-1)} \sum_{c \neq c'} \frac{\langle \mathbf{c}^{(c)} \cdot \mathbf{c}^{(c')} \rangle}{\tau_s}, \tag{9}$$

which enforces repulsion among prototypes and encourages them to capture diversity. The overall exemplar-guided prototype learning loss is therefore formulated as $\mathcal{L}_{\text{proto}} = \mathcal{L}_{\text{align}} + \mathcal{L}_{\text{disp}}$. This joint objective ensures that unlabeled samples are guided toward correct semantic directions while prototypes remain well-separated and semantically meaningful.

## 3.3 Exemplar-Guided Interpolation Consistency

Although exemplar-guided prototype learning effectively propagates class information to unlabeled data, its influence is still limited in regions near decision boundaries caused by the sparse annotation. In such ambiguous areas, prototypes alone may provide insufficient guidance, leading to unstable assignments and reduced generalization.

To address this limitation, we extend the exemplar guidance into the interpolation space. The key insight is that the influence of exemplar-anchored prototypes should also regulate interpolated samples that lie between classes. This not only diffuses exemplar semantics more broadly but also enforces decision smoothness in uncertain regions. Specifically, for a pair of unlabeled views $\mathbf{x}$ and $\mathbf{x}'$, we create a mixed view by linear interpolation

$$\mathbf{x}_m = \beta \mathbf{x} + (1 - \beta)\tilde{\mathbf{x}}', \quad \beta \sim \text{Beta}(\zeta, \zeta), \tag{10}$$

where $\tilde{\mathbf{x}}'$ is a shuffled version of $\mathbf{x}'$. $\beta \sim \text{Beta}(\zeta, \zeta)$ is a beta distribution (Zhang et al., 2018). The network takes in the mixed view and output its embedding feature $\mathbf{z}_m = \mathbf{g}(\mathbf{e}(\mathbf{x}_m)) \in \mathbb{R}^{n \times d}$. To regularize the mixed view, we introduce two complementary learning objectives from both prototype and instance perspectives.

From a prototype perspective, the distribution between mixed embedding $\mathbf{z}_m$ and constructed prototypes should be consistent with the mixed distribution. The mixed target distribution $\mathbf{p}'^{(i)}_m$ for sample $i$ is obtained by interpolating the prototype assignments of its constituent views

$$\mathbf{p}'^{(i)}_m = \beta \mathbf{p}^{(i)} + (1 - \beta)\tilde{\mathbf{p}}'^{(i)}, \tag{11}$$

where $\mathbf{p}^{(i)} = [p^{(i,1)}, p^{(i,2)}, ..., p^{(i,C)}]$ and $\tilde{\mathbf{p}}'^{(i)} = [\tilde{p}'^{(i,1)}, \tilde{p}'^{(i,2)}, ..., \tilde{p}'^{(i,C)}]$ are probability distributions of $\mathbf{x}^{(i)}$ and $\tilde{\mathbf{x}}'^{(i)}$, respectively. We optimize its distribution over the constructed prototypes to match the mixed target distribution

$$\mathcal{L}_{\text{mix-proto}} = -\frac{1}{n}\sum_{i=1}^{n}\sum_{c=1}^{C} p'^{(i,c)}_m \log p^{(i,c)}_m, \quad p^{(i,c)}_m = \frac{\exp\left(\langle \mathbf{z}^{(i)}_m \cdot \mathbf{c}^{(c)}/\tau_s \rangle\right)}{\sum_{c'=1}^{C} \exp\left(\langle \mathbf{z}^{(i)}_m \cdot \mathbf{c}^{(c')}\rangle/\tau_s\right)}. \tag{12}$$

In parallel, we encourage instance-level consistency between the mixed view and the unlabeled view. Let $\mathbf{y}$ denote the one-hot label of $\mathbf{x}$, which is pseudo-labels corresponding to the sample's index within the mini-batch. For the mixed view, the label can be calculated as $\mathbf{y}_m = \beta \mathbf{y} + (1 - \beta)\tilde{\mathbf{y}}$. We then minimize the following loss to align the similarity distribution of $\mathbf{z}_m$ with the interpolated label:

$$\mathcal{L}_{\text{mix-ins}} = -\frac{1}{n}\sum_{i=1}^{n}\sum_{j=1}^{n} \mathbf{y}^{(i,j)}_m \log \frac{\exp\left(\langle \mathbf{z}^{(i)}_m \cdot \mathbf{z}'^{(j)}\rangle/\tau_s\right)}{\sum_{j'=1}^{n} \exp\left(\langle \mathbf{z}^{(i)}_m \cdot \mathbf{z}'^{(j')}\rangle/\tau_s\right)}. \tag{13}$$

Finally, we combine the two interpolation losses into a unified objective $\mathcal{L}_{mix} = \mathcal{L}_{\text{mix-proto}} + \mathcal{L}_{\text{mix-ins}}$. Here, Eq. (12) imposes global semantic regularization by guiding prototype assignments, while Eq. (13) enforces local discrimination through instance-level consistency. Their combination provides a more holistic regularization, ensuring that exemplar guidance propagates consistently across both semantic and instance levels.

**Overall Training Objective.** The overview of our proposed method is shown in Fig. 2. The total training loss consists of three components: the base clustering alignment loss, the exemplar-guided prototype learning loss and the interpolation consistency loss. Formally, the total loss is defined as

$$\mathcal{L} = \mathcal{L}_{\text{cluster}} + \lambda \mathcal{L}_{\text{proto}} + \mu \mathcal{L}_{\text{mix}}, \tag{14}$$

where $\lambda$ and $\mu$ are positive weight coefficients that balance contributions of different loss terms.

## 4 Experiments

In this section, we first present the implementation details of the experiments. Then, we perform extensive experiments on downstream tasks across multiple benchmarks and compare our method with existing state-of-the-art methods.

Table 1: Classification top-1 accuracies of a linear and a $k$-Nearest Neighbors ($k = 5$) classifier for different datasets. All methods are based on ResNet-18 are trained for 1000 epochs on CIFAR-10/100 with a batch size of 256 and 400 epochs on ImageNet-100 with a batch size of 128.

| Method | CIFAR-10 | | CIFAR-100 | | ImageNet-100 | |
|---|---|---|---|---|---|---|
| | linear | $k$-NN | linear | $k$-NN | linear | $k$-NN |
| SimCLR (Chen et al., 2020a) | 90.74 | 85.13 | 65.78 | 53.19 | 77.64 | 65.78 |
| BYOL (Grill et al., 2020) | 92.58 | 87.40 | 70.46 | 56.46 | 80.32 | 68.94 |
| SwAV (Caron et al., 2020) | 89.17 | 84.18 | 64.88 | 53.32 | 74.28 | 63.84 |
| SimSiam (Chen & He, 2021) | 90.51 | 86.82 | 66.04 | 55.79 | 78.72 | 67.92 |
| MoCoV3 (Chen et al., 2021a) | 93.10 | 89.47 | 68.83 | 58.23 | 80.36 | 72.76 |
| W-MSE (Ermolov et al., 2021a) | 91.55 | 89.69 | 66.10 | 56.69 | 76.23 | 67.72 |
| DINO (Caron et al., 2021) | 89.52 | 86.13 | 66.76 | 56.24 | 74.92 | 64.30 |
| Barlow Twins (Zbontar et al., 2021) | 92.10 | 88.09 | 70.90 | 59.40 | 80.16 | 72.14 |
| VICReg (Bardes et al., 2022) | 92.07 | 87.38 | 68.54 | 56.32 | 79.40 | 71.94 |
| CW-RGP (Weng et al., 2022) | 92.03 | 89.67 | 67.78 | 58.24 | 76.96 | 68.46 |
| INTL (Weng et al., 2024) | 92.60 | 90.03 | 70.88 | 61.90 | 81.68 | 73.46 |
| ReSA (Weng et al., 2025) | 93.53 | 93.02 | 72.21 | 66.83 | 82.24 | 74.56 |
| **Ours** | **95.19** | **94.20** | **75.47** | **69.89** | **83.88** | **80.42** |

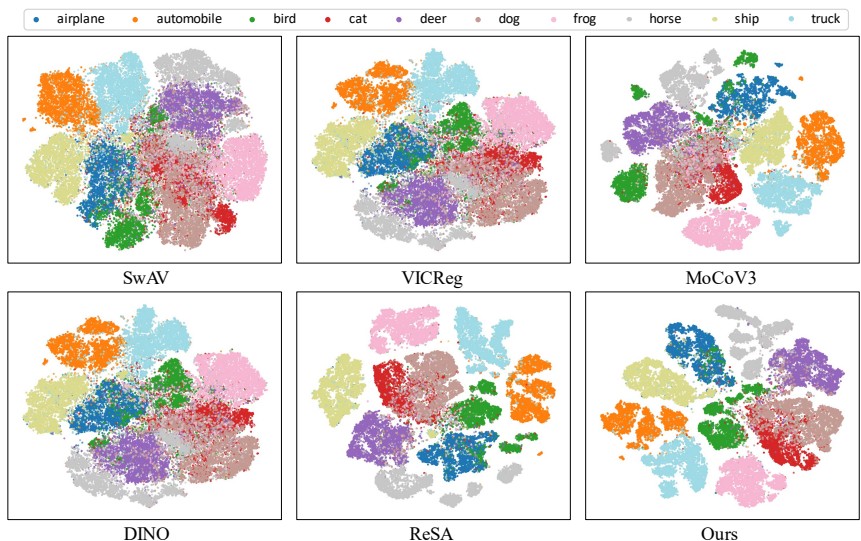

Figure 3: T-SNE visualizations of image representations on the CIFAR-10 dataset. All methods are pretrained for 1000 epochs using ResNet-18.

## 4.1 IMPLEMENTATION DETAILS

We perform pretraining on a variety of standard self-supervised learning benchmarks, including CIFAR-10/100 (Krizhevsky et al., 2009), ImageNet-100 and ImageNet-1K (Deng et al., 2009). We adopt ReSA (Weng et al., 2025) as the clustering-based method baseline, utilizing ResNets (He et al., 2016) and ViT (Dosovitskiy et al., 2021) as the encoder. During the prototype construction, we select 8 nearest neighbors for each exemplar to enrich the class-relevant feature diversity of the resulting prototype. The temperatures are fixed as $\tau_s = 0.1$ and $\tau_t = 0.04$, and the weighting factor is set to $\alpha = 0.75$. For the CIFAR datasets, all loss components are equally weighted, whereas for the ImageNet datasets, the coefficient $\mu$ is adjusted to 0.25. The sensitivity study of these hyperparameters in loss is provided in the Appendix.

## 4.2 MAIN RESULTS

In this subsection, we present experimental results on various downstream tasks. We compare our method with state-of-the-art self-supervised learning and semi-supervised learning methods. For clarity, the best results are highlighted in **bold**, while the second-best results are underlined.

Table 2: ImageNet-1K classification top-1 accuracy of a linear classifier based on a ResNet-50 encoder with various training epochs. $*$ indicates using 1% annotations (12,811 labels) during training, which is substantially higher than ours (1,000 labels).

| Method | batch size | pretrained epochs | | |
|---|---|---|---|---|
| | | 100 | 200 | 800 |
| SimCLR (Chen et al., 2020a) | 256 | 57.5 | 62.0 | 66.5 |
| | 4096 | 66.5 | 68.3 | 70.4 |
| SwAV (Caron et al., 2020) | 256 | 65.5 | 67.7 | - |
| | 4096 | 66.5 | 69.1 | 71.8 |
| MoCoV3 (Chen et al., 2021a) | 1024 | 67.4 | 71.0 | 72.4 |
| | 4096 | 68.9 | - | 73.8 |
| SimCLRv2 (Chen et al., 2020b) | 4096 | - | - | 71.7 |
| BYOL (Grill et al., 2020) | 4096 | 66.5 | 70.6 | 74.3 |
| Barlow Twins (Zbontar et al., 2021) | 2048 | 67.7 | 70.2 | 73.2 |
| VICReg (Bardes et al., 2022) | 2048 | 68.6 | 70.8 | 73.1 |
| SimSiam (Chen & He, 2021) | 256 | 68.1 | 70.0 | 71.3 |
| | 1024 | 68.0 | 69.9 | 71.1 |
| MEC (Liu et al., 2022) | 256 | 70.1 | - | - |
| | 1024 | 70.6 | 71.9 | 74.0 |
| INTL (Weng et al., 2024) | 256 | 69.5 | 71.1 | 73.1 |
| | 1024 | 69.7 | 71.2 | 73.3 |
| ReSA (Weng et al., 2025) | 256 | 71.9 | 73.4 | - |
| | 1024 | 71.3 | 73.8 | 75.2 |
| **Ours** | 256 | 72.9 | 74.6 | - |
| | 1024 | 72.8 | 74.6 | 76.4 |
| PAWS$^*$ (Assran et al., 2021) | 4096 | 72.4 | 73.5 | - |
| Suave$^*$ (Fini et al., 2023) | 256 | 72.4 | 74.0 | 75.3 |
| **Ours**$^*$ | 1024 | **73.3** | **75.0** | **76.8** |

Table 3: ImageNet classification top-1 accuracy of a linear and a $k$-Nearest Neighbors ($k = 20$) classifier based on a standard ViT-S/16 encoder. All models are pretrained for 300 epochs.

| Classifier | BYOL | SwAV | MoCoV3 | DINO | ReSA | SOP | **Ours** |
|---|---|---|---|---|---|---|---|
| linear | 71.4 | 68.5 | 72.5 | 72.5 | 72.7 | 74.3 | **74.7** |
| $k$-NN | 66.6 | 60.5 | 67.7 | 67.9 | 68.3 | 70.0 | **70.9** |

**Linear and $k$-NN Evaluation.** We first perform pretraining and evaluate classification accuracies on small and medium size datasets, i.e., CIFAR-10/100 and ImageNet100. The experimental results are shown in Tab. 1. It is obviously that our method consistently outperforms the state-of-the-art methods across all benchmarks. On CIFAR-10, we achieve 95.19% linear and 94.20% $k$-NN accuracy, outperforming ReSA by more than 1% in both metrics. On CIFAR-100, the improvements are even more pronounced, achieving 75.47% and 69.89% on linear and $k$-NN evaluation for a consistent gain of over 3% against the best competitor. Notably, our method also delivers substantial advantages on ImageNet-100, surpassing ReSA by 1.6% in linear evaluation and by almost 6% in $k$-NN classification. These results suggest that our framework is especially effective at enhancing representation quality in terms of both separability and neighborhood consistency, leading to more discriminative embeddings than prior methods. We also show the T-SNE (Van der Maaten & Hinton, 2008) visualizations of the representations learned by our proposed method and several methods on the CIFAR-10 training set in Fig. 3. These visual results corroborate our quantitative findings, showing that our framework learns semantically consistent and well-structured embeddings.

We then perform pretraining and evaluate classification accuracies on the large ImageNet-1K dataset. As shown in Tab. 2, our approach consistently surpasses existing self-supervised learning methods across different training epochs. At shorter training epochs, our method significantly outperforms competitive methods, demonstrating its superior generalization ability. When extended to longer

Table 4: Semi-supervised learning results on ImageNet-1K. All models are pretrained with ResNet-50, and then fine-tuned on 1% or 10% subset of ImageNet-1K. † indicates employing the multi-crop trick during pretraining. * indicates semi-supervised method.

| Method | top-1 | | top-5 | |
|---|---|---|---|---|
| | 1% | 10% | 1% | 10% |
| SimCLR (Chen et al., 2020a) | 48.3 | 65.6 | 75.5 | 87.8 |
| BYOL (Grill et al., 2020) | 53.2 | 68.8 | 78.4 | 89.0 |
| SwAV† (Caron et al., 2020) | 53.9 | 70.2 | 78.5 | 89.9 |
| SimCLRv2 (Chen et al., 2020b) | 57.9 | 68.4 | 82.5 | 89.9 |
| Barlow Twins (Zbontar et al., 2021) | 55.0 | 69.7 | 79.2 | 89.3 |
| VICReg (Bardes et al., 2022) | 54.8 | 69.5 | 79.4 | 89.5 |
| INTL (Weng et al., 2024) | 55.0 | 69.4 | 80.8 | 89.8 |
| ReSA (Weng et al., 2025) | 56.4 | 70.4 | 81.0 | 90.1 |
| **Ours** | **61.3** | 72.5 | **84.8** | **91.0** |
| PAWS* (Assran et al., 2021) | 59.2 | 70.2 | - | - |
| SsCL* (Zhang et al., 2022) | 60.2 | 72.1 | 82.8 | 90.9 |
| Suave* (Fini et al., 2023) | 60.4 | **72.6** | 80.6 | 90.7 |

Table 5: Transfer learning to fine-grained datasets based on ResNet-50 pretrained on ImageNet-1K. We employ a $k$-Nearest Neighbors classifier ($k = 5, 10, 20$), without requiring additional training or parameter tuning. † indicates that these methods employ the multi-crop trick.

| Method | pretrained epochs | ImageNet-1K | | | CUB-200-2011 | | | Pets-37 | | | Food-101 | | | Flowers-102 | | |
|---|---|---|---|---|---|---|---|---|---|---|---|---|---|---|---|---|
| | | 5 | 10 | 20 | 5 | 10 | 20 | 5 | 10 | 20 | 5 | 10 | 20 | 5 | 10 | 20 |
| MoCoV3 | 1000 | 67.9 | 68.9 | 68.9 | 46.8 | 48.8 | 50.4 | 85.4 | 86.5 | 86.5 | 56.3 | 58.6 | 59.7 | 83.4 | 81.6 | 80.9 |
| VICReg | 1000 | 64.3 | 65.2 | 65.6 | 33.4 | 35.4 | 36.3 | 81.5 | 82.0 | 82.3 | 56.9 | 59.6 | 61.0 | 83.4 | 83.2 | 82.6 |
| INTL | 800 | 63.6 | 64.8 | 65.1 | 26.7 | 28.0 | 29.4 | 78.4 | 79.5 | 79.7 | 55.6 | 58.1 | 59.2 | 78.8 | 77.6 | 77.2 |
| ReSA | 800 | 69.2 | 69.9 | 69.9 | 56.5 | 58.5 | 59.9 | 85.8 | 87.2 | 87.5 | 58.3 | 60.4 | 61.3 | 84.4 | 83.6 | 83.6 |
| **Ours** | 800 | **70.7** | **71.3** | **71.4** | **57.0** | **59.2** | **60.5** | **87.0** | **88.3** | **88.3** | **60.8** | **63.2** | **64.2** | 83.9 | **84.2** | **84.4** |
| SwAV† | 800 | 64.3 | 65.5 | 65.7 | 26.2 | 27.3 | 28.4 | 77.2 | 77.3 | 77.1 | 54.7 | 57.4 | 58.7 | 79.3 | 79.9 | 78.4 |
| DINO† | 800 | 66.4 | 67.4 | 67.6 | 33.8 | 35.5 | 36.8 | 81.1 | 81.6 | 80.9 | 58.2 | 60.8 | 61.8 | **84.8** | 84.1 | 83.7 |

training epochs, our method reaches 76.4%, exceeding the best reported results. Moreover, our method also outperforms existing semi-supervised learning methods that use 1% labels, further evidencing its strength in extracting rich supervision from extremely sparse annotations. We also provide a result of our method using 1% labels, and the performance of our method further improves when increasing the number of exemplars. This demonstrates that our framework can achieve strong performance with significantly fewer labeled samples than conventional semi-supervised approaches. Taken together, these findings confirm that our approach not only excels on smaller datasets but also scales robustly to large-scale settings, providing an efficient way to robust visual representations under diverse conditions.

To further validate the generality of our approach beyond convolutional backbones, we conduct experiments with a standard ViT-S/16 backbone trained on ImageNet-1K for 300 epochs. The results in Tab. 3 show that our method achieves the best performance among all considered approaches. In linear evaluation, our model attains 74.7% top-1 accuracy, surpassing other transformer-based approaches. In the $k$-NN evaluation, the advantage of our method becomes more evident, reaching 70.9% and improving upon the baseline by 2.6%. These findings demonstrate that our framework effectively adapts to transformer architectures and consistently produces robust representations.

**Semi-Supervised Learning.** We also evaluate the effectiveness of our approach in semi-supervised learning. The models are pretrained on ImageNet-1K and then fine-tuned using either 1% or 10% of the labeled data. As shown in Tab. 4, our method achieves 61.3% and 72.5% top-1 accuracy with 1% and 10% labels, respectively, consistently surpassing strong baselines such as SimCLRv2, INTL and even the recent ReSA. These improvements are also reflected in the top-5 results, where our approach reaches 84.8% and 91.0%. Our method remains competitive with dedicated semi-supervised approaches that explicitly incorporate labeled data in pretraining. The results demonstrate that our framework can achieve state-of-the-art performance with fewer labeled data.

**Transfer Learning.** We evaluate the transferability of the learned features to fine-grained classification tasks, using the ResNet-50 encoder pretrained on ImageNet-1K for 800 epochs. As summarized in Tab. 5, our method consistently outperforms existing approaches across all benchmarks. On the source dataset, our model achieves 71.4% with $k = 20$, surpassing other baselines by a clear margin. On CUB-200-2011 (Wah et al., 2011) and Oxford Pets (Parkhi et al., 2012), our approach achieves the highest accuracy, with up to 60.5% and 88.3% respectively. The benefits are even more evident on Food-101 (Bossard et al., 2014), we establish a new best result of 64.2%, exceeding ReSA by 2.9%. On Flowers-102 (Nilsback & Zisserman, 2008), our method achieves competitive performance, slightly surpassing DINO trained with multi-crop. These results highlight the robustness of our representations and their suitability for fine-grained recognition, where subtle intra-class variations and inter-class similarities make the task especially challenging.

## 5 CONCLUSION

In this paper, we introduced the One-Shot Exemplar Self-Supervised Learning, which requires only one annotated instance per class to expose the true semantic space while retaining the scalability of self-supervised learning. To solve this setting, we proposed a framework that constructs exemplar-guided prototypes augmented with discriminative neighbors, ensuring both semantic grounding and representativeness. By stabilizing prototype guidance through alignment and interpolation consistency, our method successfully transfers effective supervision from labeled exemplars to unlabeled data. Extensive experiments on CIFAR and ImageNet benchmarks demonstrated that our method enables the model to learn robust representations and achieves state-of-the-art results. However, our method relies on the assumption that each class has at least one clean exemplar available. In noisy annotation scenarios, the effectiveness of our method may be compromised. In the future, we plan to extend this setting to the noisy scenario to further optimize the self-supervised learning.

### ACKNOWLEDGMENTS

This work was supported by National Natural Science Fund of China (Nos. U24A20330, 62361166670 and 62506155), Provincial Natural Science Fund of Jiangsu (Nos. BK20251985), and Suzhou Municipal Leading Talents Fund (Nos. ZXL2025320).

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

## A  APPENDIX

### A.1  PRETRAINING SETUP AND EVALUATION PROTOCOLS

#### A.1.1  PRETRAINING SETUP

In all experiments conducted in Section 4, the pretraining setup follows ReSA (Weng et al., 2025). Specifically, the network $\mathbf{f}'$ is a momentum version of $\mathbf{f}$, where the momentum parameter follow a cosine schedule from 0.996 to 1. The projector is implemented as a three-layer MLP with a hidden layer dimension of 2048 and an output dimension of 512. The memory bank stores 65536 and 4096 embedding representations for ImageNet and CIFAR, respectively. For unlabeled images, we generate views with a weak augmentation (*ResizedCrop* and *HorizontalFlip*) and a strong augmentation (*ResizedCrop*, *HorizontalFlip*, *ColorJitter*, *Grayscale*, and *GaussianBlur*). For labeled images, only weak augmentation is applied. All optimizer-related parameters (e.g., learning rate, weight decay, and learning rate schedule) follow the same settings as ReSA.

#### A.1.2  EVALUATION PROTOCOLS

**Linear Evaluation.**  The evaluation protocol of linear evaluation on CIFAR-10/100 is the same as in ReSA (Weng et al., 2025). We freeze the pretrained encoder and train a linear classifier for 500 epochs using the Adam optimizer.

For ImageNet datasets, we also follow the same settings in ReSA. Specifically, we train the classifier for 100 epochs using SGD with a MultiStepLR scheduler, where the learning rate is decayed by a factor of 0.1 at the last 40 and 20 epochs. We use a batch size of 256 and set the weight decay to 0 during training. Differently, the learning rate on ImageNet-1K is set to 5.

**Semi-Supervised learning.**  Following the standard evaluation protocol commonly used in self-supervised learning (Chen et al., 2020b; Assran et al., 2021), we pretrain the encoder on the unlabeled ImageNet-1K dataset in a fully self-supervised manner, and then we fine-tune the pretrained encoder together with a linear classifier from the first layer of the projection head using only 1% or 10% of the labeled data. Specifically, we optimize the model for 30 epochs using SGD with a learning rate of 0.01, a batch size of 256, and a cosine decay schedule. We also compared our method against representative semi-supervised approaches (Zhang et al., 2022; Fini et al., 2023) under the same labeled data ratio. These methods are jointly trained with labeled and unlabeled data. It is worth pointing that PAWS (Assran et al., 2021) is first pretrained and then fine-tuned on labeled data.

### A.2  ADDITIONAL EXPERIMENTAL RESULTS

#### A.2.1  ABLATION STUDIES

In this section, we conduct ablation studies to evaluate the robustness of our method. All encoders are pretrained for 1000 epochs on the CIFAR-10/100 datasets with the batch size of 256.

**Effect of Training Loss.**  Tab. 6 systematically investigates the contribution of each component in our overall training objective. Using only the base clustering loss $\mathcal{L}_{\text{cluster}}$ leads to a strong baseline. Introducing the exemplar-guided prototype learning loss $\mathcal{L}_{\text{proto}}$ further improves both linear and $k$-NN evaluations, confirming that transferring the sparse supervision of labeled exemplars to unlabeled data via prototype alignment is crucial for semantically consistent representations. Incorporating the exemplar-guided interpolation consistency loss $\mathcal{L}_{\text{mix}}$ achieves the best performance, pushing the accuracy to 95.2% and 94.2% on CIFAR-10, and 75.5% and 69.9% on CIFAR-100. This validates our intuition that constraining interpolated samples encourages smoother decision boundaries and more robust prototype assignments, especially beneficial in fine-grained datasets like CIFAR-100.

**Number of Neighbor Samples.**  Tab. 7 explores the sensitivity of the number of neighbors $k$ during exemplar-guided prototype construction. Too few neighbors limits the ability to enrich prototypes, resulting in weaker generalization. Increasing $k$ to 4 and 8 progressively improves both linear and $k$-NN accuracies, with the best performance obtained at $k = 8$. This setting strikes a balance between capturing informative samples and avoiding noise. When $k$ grows further to 16, performance slightly

Table 6: Ablation studies on the different loss terms of our proposed method on CIFAR-10/100 datasets using ResNet-18 as the encoder.

| $\mathcal{L}_{\text{cluster}}$ | $\mathcal{L}_{\text{proto}}$ | $\mathcal{L}_{\text{mix}}$ | CIFAR-10 | | CIFAR-100 | |
|:---:|:---:|:---:|:---:|:---:|:---:|:---:|
| | | | linear | $k$-NN | linear | $k$-NN |
| ✓ | | | 93.5 | 93.0 | 72.2 | 66.8 |
| ✓ | ✓ | | 94.7 | 93.9 | 74.3 | 68.4 |
| ✓ | ✓ | ✓ | **95.2** | **94.2** | **75.5** | **69.9** |

declines, suggesting that incorporating too many neighbors introduces ambiguity and dilutes the discriminative power of prototypes. These findings highlight the importance of carefully controlling the neighborhood size to ensure prototypes remain both representative and robust.

Table 7: Ablation studies on the number of neighbors during exemplar-guided prototype construction.

| $k$ | CIFAR-10 | | CIFAR-100 | |
|:---:|:---:|:---:|:---:|:---:|
| | linear | $k$-NN | linear | $k$-NN |
| 2 | 94.3 | 93.4 | 74.9 | 68.3 |
| 4 | 94.6 | 93.5 | 75.0 | 68.8 |
| 8 | **95.2** | **94.2** | **75.5** | **69.9** |
| 16 | 94.5 | 93.8 | 74.6 | 68.7 |

**Effect of Discriminative Score.** Tab. 8 explores the effect of the discriminative score $\alpha$ during exemplar-guided prototype construction. The best result consistently appears at $\alpha = 0.75$. Meanwhile, the results show that incorporating the discriminative term indeed improves neighbor selection, while excessively lowering $\alpha$ diminishes its effect. This indicates that the discriminative scoring mechanism effectively enhances prototype quality when properly balanced.

Table 8: Ablation studies on the discriminative score during exemplar-guided prototype construction.

| $\alpha$ | CIFAR-10 | | CIFAR-100 | |
|:---:|:---:|:---:|:---:|:---:|
| | linear | $k$-NN | linear | $k$-NN |
| 1.00 | 94.8 | 94.0 | 75.2 | 69.1 |
| 0.75 | **95.2** | **94.2** | **75.5** | **69.9** |
| 0.50 | 95.0 | 94.0 | 75.3 | 69.3 |
| 0.25 | 94.8 | 94.0 | 75.0 | 69.1 |

**Sensitivity of Exemplars.** Our method randomly selects exemplars with a simple default random seed, and here we conduct experiments to evaluate the sensitivity to the choice of exemplars, where we select exemplars with five different random seed. As shown in Tab. 9, the performance remains highly consistent across all runs on both CIFAR-10 and CIFAR-100. This robustness arises because the exemplar is never used as a direct supervision target. Instead, its influence is mediated through neighbor aggregation and discriminative weighting, which together suppress the impact of suboptimal exemplars. Overall, the results indicate that our exemplar-guided prototype construction is stable and insensitive to random initialization.

Table 9: Sensitivity analysis of exemplar selection. We evaluate the performance of choosing exemplars with different random seeds on CIFAR-10 and CIFAR-100.

| Seed | CIFAR-10 | | CIFAR-100 | |
|:---:|:---:|:---:|:---:|:---:|
| | linear | $k$-NN | linear | $k$-NN |
| 1 | 94.8 | 93.8 | 75.6 | 68.8 |
| 2 | 95.2 | 94.1 | 75.1 | 69.0 |
| 3 | 95.1 | 93.7 | 75.9 | 69.0 |
| 4 | 94.9 | 94.0 | 74.8 | 69.7 |
| 5 | 94.9 | 94.0 | 75.0 | 68.8 |

### A.2.2 Transfer Learning on Object Detection

In this experiment, we evaluate the transferability of the learned representations on downstream dense prediction tasks. Following the standard COCO protocol (Lin et al., 2014) using Mask R-CNN, we compare our method against a wide range of self-supervised approaches pretrained on ImageNet-1K for 200 epochs with ResNet-50 backbones. As shown in Tab. 10, our approach achieves the best results across both detection and instance segmentation benchmarks. Specifically, it attains 41.3 AP in detection and 35.8 AP in segmentation, consistently outperforming all other competitive methods. The improvements are particularly evident on the $AP_{75}$ metric, where our method achieves 44.8 for detection and 38.7 for segmentation. These results indicate that our framework not only learns highly discriminative representations for classification but also transfers effectively to structured vision tasks requiring fine-grained localization and region-level consistency.

Table 10: Transfer Learning to COCO detection and instance segmentation. All competitive methods are pretrained with based on ResNet-50 on ImageNet-1K. We follow MoCo (He et al., 2020a) to apply Mask R-CNN (He et al., 2017) fine-tuned in COCO 2017 train, evaluated in COCO 2017 val. $^*$ indicates using 1% annotations (12,811 labels) during training, which is substantially higher than ours (1,000 labels).

| Method | COCO detection | | | COCO instance seg. | | |
|---|---|---|---|---|---|---|
| | $AP_{50}$ | AP | $AP_{75}$ | $AP_{50}$ | AP | $AP_{75}$ |
| Scratch | 44.0 | 26.4 | 27.8 | 46.9 | 29.3 | 30.8 |
| Supervised | 58.2 | 38.2 | 41.2 | 54.7 | 33.3 | 35.2 |
| SimCLR (Chen et al., 2020a) | 57.7 | 37.9 | 40.9 | 54.6 | 33.3 | 35.3 |
| MoCoV2 (Chen et al., 2020c) | 58.8 | 39.2 | 42.5 | 55.5 | 34.3 | 36.6 |
| BYOL (Grill et al., 2020) | 57.8 | 37.9 | 40.9 | 54.3 | 33.2 | 35.0 |
| SwAV (Caron et al., 2020) | 57.6 | 37.6 | 40.3 | 54.2 | 33.1 | 35.1 |
| SimSiam (Chen & He, 2021) | 59.3 | 39.2 | 42.1 | 56.0 | 34.4 | 36.7 |
| Barlow Twins (Zbontar et al., 2021) | 59.0 | 39.2 | 42.5 | 56.0 | 34.3 | 36.5 |
| MEC (Liu et al., 2022) | 59.8 | 39.8 | 43.2 | 56.3 | 34.7 | 36.8 |
| INTL (Weng et al., 2024) | 60.9 | 40.7 | 43.7 | 57.3 | 35.4 | 37.6 |
| ReSA (Weng et al., 2025) | 61.1 | 41.0 | 44.3 | 57.7 | 35.7 | 38.4 |
| **Ours** | **61.4** | **41.3** | **44.8** | **58.0** | **35.8** | **38.7** |
| PAWS$^*$ (Assran et al., 2021) | 60.3 | 39.9 | 42.7 | 56.6 | 34.7 | 36.7 |
| Suave$^*$ (Fini et al., 2023) | 61.0 | 40.6 | 43.9 | 57.4 | 35.2 | 37.3 |

We further validate the generality of our method on the Pascal VOC detection benchmark (Everingham et al., 2010). Following the evaluation protocol in (He et al., 2020b), we fine-tune a Faster R-CNN detector initialized from encoders pretrained on ImageNet-1K and report AP metrics on the VOC07 test set. As shown in Tab. 11, our method consistently outperforms existing compared approaches across all metrics. Combined with the COCO results, these findings demonstrate that our exemplar-guided prototype learning yields highly transferable visual features that generalize robustly across both coarse-grained and fine-grained object level tasks.

Table 11: Object detection results on Pascal VOC. All competitive methods are pretrained with based on ResNet-50 on ImageNet-1K.

| Method | $AP_{50}$ | AP | $AP_{75}$ |
|---|---|---|---|
| SimCLR (Chen et al., 2020a) | 81.8 | 55.5 | 61.4 |
| MoCoV2 (Chen et al., 2020c) | 82.3 | 57.0 | 63.3 |
| BYOL (Grill et al., 2020) | 81.4 | 55.3 | 61.1 |
| SwAV (Caron et al., 2020) | 81.5 | 55.4 | 61.4 |
| SimSiam (Chen & He, 2021) | 82.0 | 56.4 | 62.8 |
| ReSA (Weng et al., 2025) | 82.4 | 56.4 | 63.0 |
| **Ours** | **82.6** | **56.6** | **63.2** |

### A.2.3 EVALUATION ON DIVERSE DOMAINS

To evaluate the generality of our method across different domains, we further conduct experiments on both the remote sensing and medicine domains. For remote sensing, we follow the training setup reported in CMID (Muhtar et al., 2023) and evaluate various SSL methods on the UCM (UC Merced Land Use) dataset. The results in Tab. 12 show that our method achieves the best performance among all compared approaches, indicating strong cross-domain generalization despite the large appearance gap between natural and aerial imagery.

For the medical imaging domain, we evaluate the transferability of the learned representations on the ChestX-ray14 dataset. As reported in Tab. 13, our approach again achieves the highest AUC, outperforming compared SSL methods. These results demonstrate that the exemplar-guided prototype construction is not tied to a specific visual domain and can generalize effectively to scenarios with different imaging modalities and underlying data distributions.

Table 12: Comparison on the UCM testing set.

| Method | Epoch | Linear |
|---|---|---|
| BYOL (Grill et al., 2020) | 200 | 93.23 |
| Barlow (Zbontar et al., 2021) | 300 | 96.61 |
| MoCoV2 (Chen et al., 2020c) | 200 | 88.80 |
| SwAV (Caron et al., 2020) | 200 | 94.79 |
| SeCo (Manas et al., 2021) | 200 | 90.36 |
| CMID (Muhtar et al., 2023) | 200 | 96.88 |
| **Ours** | 200 | **97.99** |

Table 13: Comparison on the ChestX-ray14.

| Method | AUC |
|---|---|
| MoCoV2 (Chen et al., 2020c) | 80.46±0.54 |
| Barlow (Zbontar et al., 2021) | 80.45±0.29 |
| SimSiam (Chen & He, 2021) | 79.62±0.34 |
| SimCLRv2 (Chen et al., 2020b) | 81.23±0.09 |
| CAiD (Taher et al., 2022) | 80.72±0.29 |
| DiRA (Haghighi et al., 2022) | 81.12±0.17 |
| **Ours** | **81.62±0.42** |

### A.2.4 EVALUATION ACROSS SEMANTIC GRANULARITIES

To further examine the semantic quality of learned representations, we evaluate our method on CIFAR-100 under both fine-grained (100 classes) and coarse-grained (20 superclasses) settings. As shown in Tab. 14, our method achieves clear improvements over existing approaches across all metrics. In the fine-grained evaluation, our model reaches 75.5% linear and 69.4% $k$-NN accuracy, surpassing ReSA by more than 3% in both cases. The gains are also prominent in the coarse-grained classification task, where our approach delivers 82.7% and 82.2% accuracy under linear and $k$-NN protocols, respectively, representing a substantial leap over previous methods. These results highlight that our framework captures both fine-level distinctions and higher-level semantic structures more effectively than prior work, leading to robust performance across varying levels of label granularity.

Table 14: CIFAR-100 classification top-1 accuracy of a linear and a $k$-Nearest Neighbors ($k = 5$) classifier based on 100 fine-grained classes and 20 coarse-grained superclasses.

| Method | fine-grained | | coarse-grained | |
|---|---|---|---|---|
| | linear | $k$-NN | linear | $k$-NN |
| SimCLR (Chen et al., 2020a) | 65.8 | 53.2 | 72.5 | 67.2 |
| SwAV (Caron et al., 2020) | 64.9 | 53.3 | 70.0 | 66.3 |
| MoCoV3 (Chen et al., 2021a) | 68.8 | 58.2 | 76.4 | 68.6 |
| DINO (Caron et al., 2021) | 66.8 | 56.2 | 72.9 | 70.2 |
| VICReg (Bardes et al., 2022) | 68.5 | 56.3 | 74.3 | 69.9 |
| ReSA (Weng et al., 2025) | 72.2 | 66.8 | 79.8 | 78.8 |
| **Ours** | **75.5** | **69.4** | **82.7** | **82.2** |

### A.2.5 PARAMETRIC SENSITIVITY

We study the effect of the balancing coefficients $\lambda$ and $\mu$ in the training objective. As shown in Tab. 15, our method remains stable across a range of values on CIFAR datasets. For $\lambda$, performance steadily increases as the weight grows, with the best accuracy obtained at $\lambda = 1$. For $\mu$, small values

lead to performance degradation, but accuracy improves as the weight increases and becomes stable around $\mu = 1$, indicating the effectiveness of enforcing consistency on interpolated samples. Overall, the results suggest that our framework is not overly sensitive to precise hyper-parameter tuning, and setting all coefficients to 1 yields consistently strong performance across CIFAR datasets.

Table 15: Parametric sensitivity analysis of hyper-parameters in training loss. Classification top-1 accuracies of a linear and a $k$-Nearest Neighbors ($k = 5$) classifier for different values of hyperparameters.

| Parameter | Value | CIFAR-10 | | CIFAR-100 | |
|---|---|---|---|---|---|
| | | linear | $k$-NN | linear | $k$-NN |
| | 0.1 | 94.6 | 93.8 | 74.9 | 69.5 |
| $\lambda$ | 0.5 | 94.8 | 94.0 | 75.3 | 69.3 |
| | 1 | **95.2** | **94.2** | **75.5** | **69.9** |
| | 0.1 | 94.4 | 93.7 | 74.8 | 68.7 |
| $\mu$ | 0.5 | 94.8 | 93.8 | 75.2 | 69.4 |
| | 1 | **95.2** | **94.2** | **75.5** | **69.9** |

We further study the effect of the temperature parameters in the training objective. As shown in Tab. 16, our framework remains stable across a wide range of temperature values on both CIFAR datasets. For the student temperature, performance steadily improves as the value increases, with the best accuracy obtained at $\tau_s = 0.04$. For the teacher temperature, smaller values lead to slight degradation, while accuracy becomes consistently higher around $\tau_t = 0.1$, indicating that using a smoother teacher distribution provides more reliable training signals. Overall, these results show that the method is not overly sensitive to temperature choices.

Table 16: Parametric sensitivity analysis of temperature. Classification top-1 accuracies of a linear and a $k$-Nearest Neighbors ($k = 5$) classifier for different values of hyperparameters.

| Parameter | Value | CIFAR-10 | | CIFAR-100 | |
|---|---|---|---|---|---|
| | | linear | $k$-NN | linear | $k$-NN |
| | 0.01 | 94.9 | 94.1 | 75.2 | 69.8 |
| | 0.02 | 94.7 | 93.6 | 74.8 | 69.0 |
| $\tau_s$ | 0.03 | 94.7 | 93.8 | 75.2 | 69.2 |
| | 0.04 | **95.2** | **94.2** | **75.5** | **69.9** |
| | 0.1 | **95.2** | **94.2** | **75.5** | **69.9** |
| | 0.2 | 95.0 | 94.1 | 75.0 | 69.4 |
| $\tau_t$ | 0.3 | 94.9 | 94.1 | 74.7 | 68.9 |
| | 0.4 | 94.9 | 94.2 | 74.5 | 68.4 |

### A.2.6 ANALYSIS ON TRAINING TIME

We further compare the training efficiency of our framework with representative self-supervised methods under the same hardware and software environment. As shown in Tab. 17, our method has a computational footprint comparable to existing approaches. Specifically, the peak memory consumption of our model is slightly higher than MoCoV3, DINO, and ReSA. However, the training time per epoch remains nearly identical, indicating that the extra operations introduce negligible runtime cost. Overall, these results demonstrate that our method achieves substantial performance gains with only marginal increases in memory usage, making it both effective and efficient for large-scale self-supervised training.

### A.2.7 VISUALIZATION OF EXEMPLAR-GUIDED CLUSTERS

In Fig. 4, we show ImageNet-1K training images that are randomly chosen from clusters generated by exemplar-guided prototype construction. This visualization provides qualitative evidence for the effectiveness of our exemplar-guided prototype construction in shaping a semantically meaningful representation space, validating that our prototype construction builds a coherent foundation for guiding the overall representation learning.

Table 17: Comparison of computational overhead among various SSL methods. All methods are based on ResNet-50 pretrained on ImageNet-1K with the batch size of 1024. We perform peak memory (GB per GPU) and training time (hours per epoch) on the same environment and machine equipped with 8 V100-32GB GPUs using 32 dataloading workers under mixed-precision.

| Method | memory | time |
|--------|--------|------|
| MoCoV3 | 14.9 | 0.39 |
| DINO | 15.0 | 0.41 |
| ReSA | **14.8** | **0.36** |
| Ours | 15.8 | 0.40 |

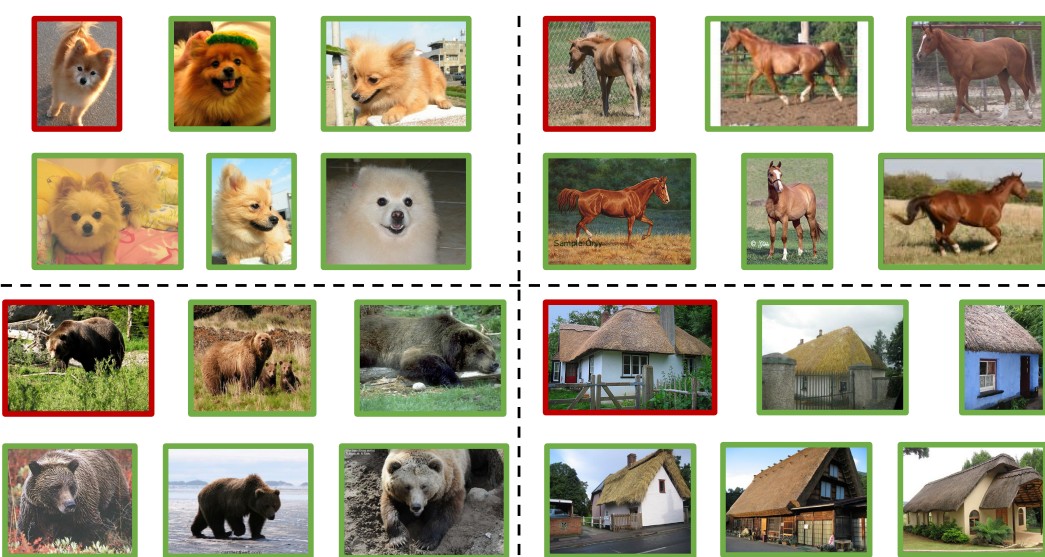

Figure 4: Visualization of constructed exemplar-guided prototypes. The Red boarder marks the labeled exemplars. The Green boarder marks the discriminative neighbors.

### A.3 THE USE OF LARGE LANGUAGE MODELS

In the preparation of this manuscript, large language models (LLMs) were used solely as a tool to aid and polish the writing. The specific use cases were limited to grammar and syntax correction, sentence polishing. The LLMs were used under the guidance and with the critical oversight of the human authors to ensure the integrity and accuracy of the academic work.

