# OpenReview forum: "One-Shot Exemplars for Class Grounding in Self-Supervised Learning"
_ICLR.cc/2026/Conference — ICLR 2026 Poster_

### Official Review · Reviewer_Bjyr · 2025-10-16

**Soundness:** 3
**Presentation:** 4
**Contribution:** 3
**Rating:** 8
**Confidence:** 4

**Summary:**

This paper focuses on one-shot exemplar self-supervised learning, a new problem to self-supervised learning that uses one labeled example per class. To address this issue, the paper incorporates a prototype learning algorithm and an interpolation consistency module. Extensive experimental results validate the effectiveness of the proposed method.

**Strengths:**

The paper is well-written.

The problem of self-supervised learning with one-shot exemplars is interesting and novel to the literature. Utilizing the available supervision information for self-supervised learning is a critical research problem. Therefore, it will create new opportunities in the field.

The proposed overall algorithm is simple yet effective. The different components of the algorithm are effective and reasonable. The design of the algorithms for exemplar-guided prototype construction, exemplar-guided prototype learning, and exemplar-guided interpolation consistency is coherent and interesting.

Extensive experiments validate the effectiveness of the proposed method.

**Weaknesses:**

Is the proposed approach sensitive to the choice of the selected example for each class? Since one exemplar is very few, will it cause big variations on the algorithm performance?

Since there are different losses for the proposed methods, it would be beneficial to conduct ablation studies to confirm the effectiveness of each component, although the experimental results are already very comprehensive.

**Questions:**

I have no major questions on this paper.

---

> ### Author Response · Authors · 2025-11-19
>
> Thank you for your appreciation of the novelty and experimental results of our paper! Thanks also for your very insightful and constructive suggestions! Our point-by-point responses are as follows.
>
> ---
>
> **Comment_1:** Sensitivity to the choice of exemplars.
>
> **Response_1:** Thanks for your comment. In fact, our method randomly selects exemplars with a simple default random seed, and here we want to conduct additional experiments to evaluate the sensitivity to the choice of exemplars.
> We repeat the exemplar selection with five different random seeds. The downstream accuracies are reported below.
> | Seed | CIFAR-10 (Linear) | CIFAR-10 (k-NN) | CIFAR-100 (Linear) | CIFAR-100 (k-NN) |
> | :-: | :-: | :-: | :-: | :-: |
> |1|94.8|93.8|75.6|68.8|
> |2|95.2|94.1|75.1|69.0|
> |3|95.1|93.7|75.9|69.0|
> |4|94.9|94.0|74.8|69.7|
> |5|94.9|94.0|75.0|68.8|
>
> The similar performance across different seeds shows that our method is highly robust to exemplar choice, confirming that the representation is not overly dependent on selecting a particularly good exemplar.
>
> ---
>
> **Comment_2:** Ablation studies.
>
> **Response_2:** Thanks for your comment. The ablation studies were included in the Appendix (Sec. A.2.1). Here we want to list them again for clear explanations, where we evaluate the influences of each loss term and the number of neighbors during prototype construction. The results are reported below.
> | $\mathcal{L}_\text{cluster}$ | $\mathcal{L}_\text{proto}$ | $\mathcal{L}_\text{mix}$ | CIFAR-10 (Linear) | CIFAR-10 (k-NN) | CIFAR-100 (Linear) | CIFAR-100 (k-NN) |
> |----|----|----|:------:|:------:|:------:|:------:|
> | &checkmark; | | |93.5|93.0|72.2|66.8|
> | &checkmark; | &checkmark; | |94.7|93.9|74.3|68.4|
> | &checkmark; | &checkmark; | &checkmark; |95.2|94.2|75.5|69.9|
>
> Removing any of these terms consistently degrades performance, confirming that all loss components contribute to the final representation quality.
>
> | $k$ | CIFAR-10 (Linear) | CIFAR-10 (k-NN) | CIFAR-100 (Linear) | CIFAR-100 (k-NN) |
> | :-: | :-: | :-: | :-: | :-: |
> |2|94.3|93.4|74.9|68.3|
> |4|94.6|93.5|75.0|68.8|
> |8|95.2|94.2|75.5|69.9|
> |16|94.5|93.8|74.6|68.7|
>
> Increasing the number of neighbors from consistently improves performance, with $k=8$ yielding the best results by providing sufficiently enriched yet clean prototype estimates. Using too many neighbors slightly degrades accuracy, suggesting that overly large neighborhoods introduce noise and reduce prototype discriminability.

---

> ### Comment · Reviewer_Bjyr · 2025-11-21
> **Thanks for the rebuttal!**
>
> Thanks for the rebuttal. I am satisfied with the rebuttal. I will maintain my score to accept this submission. Please include the new experimental results in the revised version of the paper.

---

> > ### Author Response · Authors · 2025-11-27
> >
> > Dear reviewer Bjyr,
> >
> > Thank you very much for your positive feedback. We sincerely appreciate your constructive comments. We have updated the revised paper including the new experimental results as requested. Thank you once again for your thoughtful review and valuable insights.
> >
> > Best regards,
> >
> > The Authors of Paper ID-6489

---

### Official Review · Reviewer_Ubdk · 2025-10-30

**Soundness:** 4
**Presentation:** 3
**Contribution:** 3
**Rating:** 8
**Confidence:** 4

**Summary:**

This paper introduces a novel and practical learning setting called One-Shot Exemplar Self-Supervised Learning (OSESSL). The core motivation is derived from that self-supervised learning (SSL) fails to specify the class space, which inevitably weakens the effectiveness of the learned representations for downstream tasks with intrinsic class structures. Instead of relying on self-generated signals, the authors propose using a single annotated instance per class to provide minimal yet crucial supervision, guiding the model toward meaningful semantic structures while retaining the scalability of SSL. The central mechanism to tackle the new setting is the "Exemplar-Guided Prototype Construction," where each single-labeled exemplar is used to build a robust class prototype. This is achieved not by using the exemplar in isolation, but by enriching it with its most discriminative neighbors from the vast pool of unlabeled data. These prototypes then provide semantic guidance for a clustering-based SSL objective. The framework is further enhanced with a prototype dispersion loss to prevent collapse and an "Exemplar-Guided Interpolation Consistency" loss, which regularizes the model on mixed samples to improve decision boundary robustness. Extensive experimental results on standard benchmarks demonstrates that the proposed method significantly outperforms a wide array of state-of-the-art SSL methods across multiple evaluation protocols and demonstrates strong performance on both CNN and Vision Transformer backbones.

**Strengths:**

1. The introduction of the OSESSL setting is a significant contribution in itself. The authors provide a very lucid argument for its necessity. Unlike traditional SSL that lacks explicit semantic grounding, OSESSL leverages a single labeled exemplar per class to guide representation learning in a scalable manner. The OSESSL setting offers semantic grounding with a truly negligible annotation cost (a complexity of $\mathcal{O}(1)$ with respect to dataset size, as the authors rightly point out). This is an extremely practical scenario for many real-world applications where identifying and labeling one canonical example of each class is far more feasible than labeling thousands of instances.

2. The proposed method is simple yet effective, combining exemplar-guided prototype alignment, prototype dispersion, and interpolation consistency to jointly promote discriminative and stable representation learning. The design is also supported by theoretical reasoning. The method is evaluated across diverse datasets, and significant performance makes the empirical evidence convincing. Ablation studies further confirm the complementary effects of the proposed components and the stability of key hyperparameters.

3. The paper is clearly written and well-structured, making it easy to follow. The authors effectively present the motivation, identify the limitations of prior work, and justify their proposed approach. Figures and tables are well-designed to illustrate key results, and the appendices provide useful supplementary details that enhance the overall clarity and completeness of the work.

**Weaknesses:**

1. The neighbor selection mechanism is key to the method's success, as it relies on a meaningful feature space to identify semantically similar instances. However, in the early stages of training, the encoder is not yet powerful, and the feature space is likely to be poorly structured. I concern that the initial neighbor selections could be noisy or incorrect, reinforcing incorrect associations early on.

2. Lack of ablation studies of exemplar-guided prototype construction. It would be helpful to analyze how the choice of exemplars influences performance and to evaluate the sensitivity or benefit of the discriminative-neighbor weighting parameter. Such studies would clarify the robustness and generality of the proposed mechanism.

**Questions:**

Have the authors evaluated the robustness of OSESSL under exemplar noise, or considered mechanisms to mitigate its effects? If such analysis has not been conducted, what challenges or design choices would be most critical for extending the method to noisy one-shot supervision?

---

> ### Author Response · Authors · 2025-11-19
> **Response to Reviewer Ubdk (1/2)**
>
> We thank the reviewer for the insightful feedback and positive comments of our work's motivation and experimental results. We provide a point-by-point response to the comments raised below.
>
> ---
>
> **Comment_1:** Clarification of neighbor selection mechanism.
>
> **Response_1:** Thanks for your comment. Indeed, the encoder’s representations are less reliable during the early stages of training. Our method mitigates this in several ways:
> 1. Prototypes are computed at every iteration using the updated memory bank. Thus, even if early neighbors are imperfect, their influence decays rapidly as the representation space improves. This ensures that prototypes converge toward the correct class structure over training.
> 2. The discriminative score does not rely solely on intra-class similarity. By explicitly penalizing high similarity to exemplars of other classes, it filters out ambiguous or misaligned candidates. This makes selection more robust to early-stage feature noise.
> 3. Rather than using hard nearest neighbors, we use soft weights normalized over the memory bank. Even if early neighbors are noisy, their influence is limited, preventing error reinforcement.
>
> ---
>
> **Comment_2:** Ablation studies of exemplar-guided prototype construction.
>
> **Response_2:** Thanks for your comment. We want to follow the reviewer’s suggestion to conduct ablation studies of exemplar-guided prototype construction to evaluate the robustness.
> We first analyze the sensitivity to the choice of exemplars. In fact, our method randomly selects exemplars with a simple default random seed. Here we repeat the exemplar selection with five different random seeds. The downstream accuracies are reported below.
> | Seed | CIFAR-10 (Linear) | CIFAR-10 (k-NN) | CIFAR-100 (Linear) | CIFAR-100 (k-NN) |
> | :-: | :-: | :-: | :-: | :-: |
> |1|94.8|93.8|75.6|68.8|
> |2|95.2|94.1|75.1|69.0|
> |3|95.1|93.7|75.9|69.0|
> |4|94.9|94.0|74.8|69.7|
> |5|94.9|94.0|75.0|68.8|
>
> The similar performance across different seeds shows that our method is highly robust to exemplar choice, confirming that the representations are not overly dependent on selecting a particularly good exemplar.
> We also investigate the parametric sensitivity of discriminative score $\alpha$. The results are reported below.
> | $\alpha$ | CIFAR-10 (Linear) | CIFAR-10 (k-NN) | CIFAR-100 (Linear) | CIFAR-100 (k-NN) |
> |--|:-:|:-:|:-:|:-:|
> |1.00|94.8|94.0|75.2|69.1|
> |0.75|95.2|94.2|75.5|69.9|
> |0.50|95.0|94.0|75.3|69.3|
> |0.25|94.8|94.0|75.0|69.1|
>
> The best result consistently appears at $\alpha=0.75$. Meanwhile, the results show that incorporating the discriminative term indeed improves neighbor selection, while excessively lowering $\alpha$ diminishes its effect. This indicates that the discriminative scoring mechanism effectively enhances prototype quality when properly balanced.

---

> > ### Author Response · Authors · 2025-11-19
> > **Response to Reviewer Ubdk (2/2)**
> >
> > **Comment_3:** Analysis on noisy exemplars.
> >
> > **Response_3:** Thanks for your comment. While our setting follows the standard assumption of one clean exemplar per class, we agree with the reviewer that examining robustness under exemplar noise is also valuable. Therefore, we conduct additional experiments on CIFAR-10 and CIFAR-100 where we artificially corrupt exemplar labels with two types of noise: Permutation (a portion of exemplars swap labels among the class set) and Random (a portion of exemplars are assigned random labels from the full label space). We vary the noise ratio to 20%, 50%, and 80%. The results are reported below.
> > | Type | Ratio | CIFAR-10 (Linear) | CIFAR-10 (k-NN) | CIFAR-100 (Linear) | CIFAR-100 (k-NN) |
> > | --- | --- | :-: | :-: | :-: | :-: |
> > | Permutation | 20% | 95.1 | 94.2 | 75.3 | 68.6 |
> > | | 50% | 94.7 | 94.0 | 74.9 | 69.0 |
> > | | 80% | 95.3 | 94.0 | 75.5 | 69.2 |
> > | Random | 20% | 94.8 | 93.9 | 74.9 | 68.8 |
> > | | 50% | 94.6 | 93.5 | 74.6 | 67.6 |
> > | | 80% | 94.2 | 93.3 | 74.3 | 67.2 |
> >
> > As our method does not directly rely on exemplar labels for supervision, and those exemplars act as feature anchors, all prototypes are actually formed through neighbor-based aggregation. Accordingly, from the above results, we can easily observe that exemplar embeddings remain aligned with their semantic neighbors under permutation noise, leading to high robustness. In contrast, random noise breaks this alignment by assigning labels that are inconsistent with the exemplar's feature cluster, thus resulting in the corresponding performance drop. Meanwhile, we also compare our method with PAWS and Suave under the challenging random noise on CIFAR-10, and the results are presented below.
> > | Method | Ratio | CIFAR-10 (Linear) | CIFAR-10 (k-NN) |
> > | --- | --- | :-: | :-: |
> > | PAWS | 20% | 92.8 | 89.5 |
> > | | 50% | 92.1 | 88.1 |
> > | | 80% | 90.2 | 87.1 |
> > | Suave | 20% | 90.1 | 82.4 |
> > | | 50% | 88.2 | 80.7 |
> > | | 80% | 86.2 | 78.7 |
> > | Ours | 20% | 94.8 | 93.9 |
> > | | 50% | 94.6 | 93.5 |
> > | | 80% | 94.2 | 93.3 |
> >
> > Under random noise corruption, both baselines exhibit substantially larger degradation than our method, where our method degrades much more mildly in all cases. This clearly demonstrates that our exemplar-guided prototype construction and discriminative weighting effectively dilute the impacts of noisy exemplars, thereby making the method inherently more robust.

---

### Official Review · Reviewer_dQe4 · 2025-11-01

**Soundness:** 3
**Presentation:** 3
**Contribution:** 2
**Rating:** 4
**Confidence:** 3

**Summary:**

The paper addressed self-supervised learning, but with a new paradigm called One-Shot Exemplar Self-Supervised Learning (OSESSL). In OSESSL, the challenge is to use the sparse supervision to guide representation learning. First, the authors build class-specific prototypes and use the prototypes to guide representation learning. Moreover, the authors proposed an interpolation consistency loss to provide the regularization and improve the decision boundaries. The authors have conducted extensive experiments to validate the effectiveness of the  proposed method in various settings, such as SSL, semi-supervised learning and transfer learning.

**Strengths:**

1. The paper is well motivated

Considering representation learning in a semantically grounded manner is likely to encourage the model to learn more informative representations.

2. The presentation is clear

3. The paper is easy to follow

**Weaknesses:**

1. The proposed OSESSL setting seems unnecessary

2. The proposed exemplar-guided alignment seems similar to PAWS

3. The ablation study is missing

4. The semi-supervised learning setting is strange

Please see the Question section below for details.

**Questions:**

1. The proposed OSESSL setting seems unnecessary

While the authors claim this OSESSL as one of their contributions, I am not convinced that OSESSL brings new insights to the community. Its entire definition is covered by semi-supervised learning; the one-exemplar assumption is still under the few-shot semi-supervised learning paradigm. Having one or 3 samples does not change the nature of sparse supervision.

2. The proposed exemplar-guided alignment seems similar to PAWS

I would love to see a direct comparison between the proposed exemplar-guided alignment with PAWS. The major difference is that PAWS uses labeled data as anchors, and here the authors use the reconstructed exemplars. Under such sparse supervision, I am not sure how much more information the reconstructed exemplar gathers, as it has to stay close to the labeled data to avoid the loss of semantic meaning. Therefore, I find these two methods very similar.

3. The ablation study is missing

While the ablation study is promised in line 312 in the experiment section, there is no ablation study provided in this section, making it hard to validate the effectiveness of the proposed method.

4. The semi-supervised learning setting is strange

I find the semi-supervised learning setting in line 425 very strange. It is more like a transfer learning rather than semi-supervised learning, if the model is first trained in SSL on ImageNet-1k, and then fine-tuned on labeled data. In general, semi-supervised learning provides both labeled and unlabeled data at the same time. I hope the authors could elaborate on the reason for setting up the experiments in this way.

**Details Of Ethics Concerns:**

No concern.

---

> ### Author Response · Authors · 2025-11-19
> **Response to Reviewer dQe4 (1/2)**
>
> We appreciate the reviewer's comments for our paper. We address the concerns raised point by point below.
>
> ---
>
> **Comment_1:** The contribution of proposed setting.
>
> **Response_1:** Thanks for your comment. We would like to clarify below why the OSESSL setting is both necessary and distinct from conventional semi-supervised/few-shot learning paradigms.
>
> **1. A necessary thing to avoid “no-free-lunch”.** Self-supervised learning learns representations driven by pretext tasks, which are not guaranteed to align with semantic class structures. OSESSL directly addresses this issue by introducing the minimal possible supervision required to ground the representation learning process towards the class space. This minimal supervision is required to overcome the “no-free-lunch” limitation of self-supervised learning.
>
> **2. Different learning goals.** In a general form, although the labeling scenario of one exemplar per class is a special case in terms of label quantity, OSESSL differs from standard semi-supervised learning in its learning purpose and mechanism. Semi-supervised learning uses labels to directly train classifier boundaries, and therefore the labels themselves dominate how the model organizes classes. In contrast, OSESSL never predicts labels nor uses label values explicitly. Instead, exemplars solely indicate inter-class distinctness and serve as semantic anchors that shape the representation space, while the unlabeled data drive the main optimization. Thus, OSESSL is not a conventional semi-supervised learning case with fewer labels, but a distinct problem setting aimed at understanding how minimal semantic supervision interacts with self-supervised representation learning.
>
> **3. Different training and test settings.** Few-shot methods focus on learning to classify unseen categories. OSESSL aims to study how a single exemplar per class can structurally guide self-supervised representation learning. Therefore, they are two completely different types of learning paradigms.
>
> ---
>
> **Comment_2:** Difference with PAWS.
>
> **Response_2:** Thanks for your comment. While both PAWS and our method leverage limited supervision as anchors, our method differs conceptually and technically in three key aspects:
> 1. Our method introduces dynamic prototype construction. By aggregating information from the most discriminative neighbors, we form a more stable and representative class centroid. This is analogous to creating a "multi-view" representation of the class from the unlabeled data. PAWS employs a static support mechanism, utilizing the raw labeled data directly as the anchor, which may suffer the inherent bias of a single instance.
> 2. Our method explicitly maximizes the separation between class prototypes through a dispersive loss. This shapes the representation space so that the prototypes of different classes remain well-separated, which is especially important for the dynamic prototype in the one-shot annotation scenario. PAWS does not impose any constraint that pushes the feature centroids of different classes.
> 3. Our method explicitly handles ambiguous or boundary samples through feature interpolation. This allows smooth transitions between nearby classes and encourages representation continuity across class boundaries. PAWS lacks such a mechanism and can hardly smoothen representation space around difficult samples.

---

> > ### Author Response · Authors · 2025-11-19
> > **Response to Reviewer dQe4 (2/2)**
> >
> > **Comment_3:** Missing of ablation studies.
> >
> > **Response_3:** Thanks for your comment. The ablation studies were included in the Appendix (Sec. A.2.1), but due to layout adjustments, we mistakenly left a reference at line-312 pointing to the main experiment section. Here we want to list them again for clear explanations, where we evaluate the influences of each loss term and the number of neighbors during prototype construction. The results are reported below.
> > | $\mathcal{L}_\text{cluster}$ | $\mathcal{L}_\text{proto}$ | $\mathcal{L}_\text{mix}$ | CIFAR-10 (Linear) | CIFAR-10 (k-NN) | CIFAR-100 (Linear) | CIFAR-100 (k-NN) |
> > |----|----|----|:------:|:------:|:------:|:------:|
> > | &checkmark; | | |93.5|93.0|72.2|66.8|
> > | &checkmark; | &checkmark; | |94.7|93.9|74.3|68.4|
> > | &checkmark; | &checkmark; | &checkmark; |95.2|94.2|75.5|69.9|
> >
> > Removing any of these terms consistently degrades performance, confirming that all loss components contribute to the final representation quality.
> >
> > | $k$ | CIFAR-10 (Linear) | CIFAR-10 (k-NN) | CIFAR-100 (Linear) | CIFAR-100 (k-NN) |
> > | :-: | :-: | :-: | :-: | :-: |
> > |2|94.3|93.4|74.9|68.3|
> > |4|94.6|93.5|75.0|68.8|
> > |8|95.2|94.2|75.5|69.9|
> > |16|94.5|93.8|74.6|68.7|
> >
> > Increasing the number of neighbors consistently improves performance, with $k=8$ yielding the best results by providing sufficiently enriched yet clean prototype estimates. Using too many neighbors slightly degrades accuracy, suggesting that overly large neighborhoods introduce noise and reduce prototype discriminability.
> > We will correct the misplaced reference in the revised version.
> >
> > ---
> >
> > **Comment_4:** The setting of semi-supervised learning.
> >
> > **Response_4:** Thanks for your comment. Our semi-supervised evaluation follows the standard protocol used in self-supervised learning (e.g., SimCLR, MoCo, BYOL, SwAV). In this setup, the model is first trained (with a self-supervised manner) on the unlabeled ImageNet-1K dataset, and then fine-tuned on a subset of labeled data (e.g., 1%, 10%) for evaluation. This two-stage procedure is widely adopted to evaluate the transferability of self-supervised representations. This differs from semi-supervised methods, which are jointly trained with unlabeled and labeled data. We also compared our method against representative semi-supervised approaches under the same labeled data ratio, demonstrating competitive or superior performance.
> >
> > We will further clarify the above semi-supervised learning setup in the revised version.

---

### Official Review · Reviewer_P5Pb · 2025-11-02

**Soundness:** 2
**Presentation:** 2
**Contribution:** 2
**Rating:** 4
**Confidence:** 1

**Summary:**

This paper introduces a new setting called One-Shot Exemplar Self-Supervised Learning (OSESSL), where self-supervised learning is augmented with just one labeled example per class to ground the representations in the true class space. The authors argue that this minimal supervision—essentially O(1) annotation cost relative to dataset size—can significantly boost downstream performance without losing SSL's scalability. They propose a framework that builds class-specific prototypes from the exemplar plus selected unlabeled neighbors, enforces alignment across views, adds dispersion to prevent collapse, and includes an interpolation consistency regularization for robustness near decision boundaries. Experiments on CIFAR-10/100, ImageNet-100/1K show consistent gains over SSL baselines like ReSA, DINO, etc., with notable improvements in k-NN (e.g., +6% on ImageNet-100) and linear classification. Transfer to semi-supervised, detection, and fine-grained tasks also looks strong

**Strengths:**

1. The OSESSL idea is clever and addresses a real gap in SSL—lack of class grounding—while keeping annotations negligible. It's a nice middle ground between pure SSL and semi-supervised learning, especially since class count grows slower than data volume in big datasets like LAION. This could inspire more work on "minimal supervision" hybrids.
2.The prototype construction (using discriminative scores for neighbors) and interpolation consistency are straightforward extensions to clustering-based SSL. The math derivations (e.g., gradient analysis of alignment loss) provide good intuition on why it works. No overly complex bells and whistles, which makes it reproducible.
3

**Weaknesses:**

1.The method relies on one high-quality exemplar per class, but real-world data often has noise or ambiguity. The conclusion mentions extending to noisy scenarios, but no experiments here—would be good to test robustness with mislabeled or atypical exemplars.
2. While gains over pure SSL are clear, the semi-sup baselines (PAWS, Suave) use 1% labels (~12k on ImageNet), which is 12x more than yours (1k classes = 1k labels). Claiming superiority feels a bit stretched without ablating how performance scales with more exemplars. Also, some citations are to 2025 papers (ReSA, SOP)—fine for anon review, but ensure they're public.
3. Mostly image classification-focused; more on non-vision (e.g., if applicable) or diverse domains (medical, satellite) would strengthen generality. Detection results are good, but only on COCO—VOC or others? Also, no analysis on failure cases, like classes with high intra-variance.
4. Appendix shows stability, but temperatures τs/τt fixed at 0.1/0.04—why these? And α=0.75 in discriminative score; a sweep there might reveal more.

**Questions:**

See Weaknesses

---

> ### Author Response · Authors · 2025-11-19
> **Response to Reviewer P5Pb (1/2)**
>
> We appreciate the reviewer's comments for our paper. We address the concerns raised point by point below.
>
> ---
>
> **Comment_1:** Analysis on noisy exemplars.
>
> **Response_1:** Thanks for your comment. While our setting follows the standard assumption of one clean exemplar per class, we agree with the reviewer that examining robustness under exemplar noise is also valuable. Therefore, we conduct additional experiments on CIFAR-10 and CIFAR-100 where we artificially corrupt exemplar labels with two types of noise: Permutation (a portion of exemplars swap labels among the class set) and Random (a portion of exemplars are assigned random labels from the full label space). We vary the noise ratio to 20%, 50%, and 80%. The results are reported below.
> | Type | Ratio | CIFAR-10 (Linear) | CIFAR-10 (k-NN) | CIFAR-100 (Linear) | CIFAR-100 (k-NN) |
> | --- | --- | :-: | :-: | :-: | :-: |
> | Permutation | 20% | 95.1 | 94.2 | 75.3 | 68.6 |
> | | 50% | 94.7 | 94.0 | 74.9 | 69.0 |
> | | 80% | 95.3 | 94.0 | 75.5 | 69.2 |
> | Random | 20% | 94.8 | 93.9 | 74.9 | 68.8 |
> | | 50% | 94.6 | 93.5 | 74.6 | 67.6 |
> | | 80% | 94.2 | 93.3 | 74.3 | 67.2 |
>
> As our method does not directly rely on exemplar labels for supervision, and those exemplars act as feature anchors, all prototypes are actually formed through neighbor-based aggregation. Accordingly, from the above results, we can easily observe that exemplar embeddings remain aligned with their semantic neighbors under permutation noise, leading to high robustness. In contrast, random noise breaks this alignment by assigning labels that are inconsistent with the exemplar's feature cluster, thus resulting in the corresponding performance drop. Meanwhile, we also compare our method with PAWS and Suave under the challenging random noise on CIFAR-10, and the results are presented below.
> | Method | Ratio | CIFAR-10 (Linear) | CIFAR-10 (k-NN) |
> | --- | --- | :-: | :-: |
> | PAWS | 20% | 92.8 | 89.5 |
> | | 50% | 92.1 | 88.1 |
> | | 80% | 90.2 | 87.1 |
> | Suave | 20% | 90.1 | 82.4 |
> | | 50% | 88.2 | 80.7 |
> | | 80% | 86.2 | 78.7 |
> | Ours | 20% | 94.8 | 93.9 |
> | | 50% | 94.6 | 93.5 |
> | | 80% | 94.2 | 93.3 |
>
> Under random noise corruption, both baselines exhibit substantially larger degradation than our method, where our method degrades much more mildly in all cases. This clearly demonstrates that our exemplar-guided prototype construction and discriminative weighting effectively dilute the impacts of noisy exemplars, thereby making the method inherently more robust.
>
> ---
>
> **Comment_2:** Scaling with more exemplars.
>
> **Response_2:** Thanks for your comment. To evaluate how performance scales with more exemplars, we additionally train our model on ImageNet-1K with 1% labeled data. The results are presented below.
> | Method | 100-epochs | 200-epochs | 800-epochs |
> |---|:--:|:--:|:--:|
> | PAWS (1 percent) | 72.4 | 73.5 | - |
> | Suave (1 percent) | 72.4 | 74.0 | 75.3 |
> | Ours (1 sample) | 72.8 | 74.6 | 76.4 |
> | Ours (1 percent) | **73.3** | **75.0** | **76.8** |
>
> The performance of our method under different training epochs consistently improves when increasing the number of exemplars, and the gains keep stable across training epochs. These results successfully validate that our method scales smoothly with additional supervision.
>
> Additionally, all 2025’s papers cited in our paper are publicly available.

---

> > ### Author Response · Authors · 2025-11-19
> > **Response to Reviewer P5Pb (2/2)**
> >
> > **Comment_3:** Generality with diverse domains and other tasks.
> >
> > **Response_3:** Thanks for your comment. To evaluate the generality of our method across different domains and tasks, we further conduct additional experiments on remote sensing, medicine, and VOC detection. Specifically, we evaluate the performance of various SSL methods on the UCM (UC Merced land use) testing set. We follow the training epochs and results reported in CMID [R1]. In the medical domain, we evaluate the capacity of transfer learning on the ChestX-ray14 dataset. The results for remote sensing, medicine, and VOC detection are reported in sequence below.
> > | Method | epoch | linear |
> > |---|---|---|
> > | BYOL | 200 | 93.23 |
> > | Barlow Twins | 300 | 96.61 |
> > | MoCoV2 | 200 | 88.80 |
> > | SwAV | 200 | 94.79 |
> > | SeCo | 200 | 90.36 |
> > | CMID | 200 | 96.88 |
> > | Ours | 200 | **97.99** |
> >
> > | Method | AUC |
> > |---|---|
> > | MoCoV2 | 80.46±0.54 |
> > | Barlow Twins | 80.45±0.29 |
> > | SimSiam | 79.62±0.34 |
> > | SimCLRv2 | 81.23±0.09 |
> > | CAiD | 80.72±0.29 |
> > | DiRA | 81.12±0.17 |
> > | Ours | **81.62±0.42** |
> >
> > | Method | AP50 | AP | AP75 |
> > |---|---|---|---|
> > | SimCLR | 81.8 | 55.5 | 61.4 |
> > | MoCoV2 | 82.3 | 57.0 | 63.3 |
> > | BYOL | 81.4 | 55.3 | 61.1 |
> > | SwAV | 81.5 | 55.4 | 61.4 |
> > | SimSiam | 82.0 | 56.4 | 62.8 |
> > | ReSA | 82.4 | 56.4 | 63.0 |
> > | Ours | **82.6** | **56.6** | **63.2** |
> >
> > These results show that the method generalizes well to remote sensing, medicine, and VOC detection, thereby suggesting that the exemplar-guided prototype representation is actually not a domain-specific technique.
> >
> > High intra-class variance is indeed a challenging case, as the single exemplar may not fully capture class diversity. However, our method does not rely solely on the exemplar itself, but aggregates multiple semantically consistent neighbors around it to form the prototype representation. This dynamic aggregation allows diverse intra-class instances to contribute to the class structure, reducing sensitivity to exemplar bias. Empirically, our fine-grained recognition results (Tab. 5 and Tab. 9) already demonstrate robustness in those datasets with large intra-class variability, confirming the model’s ability to maintain stable representations under high intra-class variance.
> >
> > ---
> >
> > **Comment_4:** Selection of temperature and discriminative score.
> >
> > **Response_4:** Thanks for your comment. We investigate the parametric sensitivity of $\tau_t$, $\tau_s$ and $\alpha$ to validate their optimal settings.
> > | $\tau_t$ | CIFAR-10 (Linear) | CIFAR-10 (k-NN) | CIFAR-100 (Linear) | CIFAR-100 (k-NN) |
> > |---|:-:|:-:|:-:|:-:|
> > |0.01|94.9|94.1|75.2|69.8|
> > |0.02|94.7|93.6|74.8|69.0|
> > |0.03|94.7|93.8|75.2|69.2|
> > |0.04|95.2|94.2|75.5|69.9|
> >
> > | $\tau_s$ | CIFAR-10 (Linear) | CIFAR-10 (k-NN) | CIFAR-100 (Linear) | CIFAR-100 (k-NN) |
> > |--|:-:|:-:|:-:|:-:|
> > |0.1|95.2|94.2|75.5|69.9|
> > |0.2|95.0|94.1|75.0|69.4|
> > |0.3|94.9|94.1|74.7|68.9|
> > |0.4|94.9|94.2|74.5|68.4|
> >
> > | $\alpha$ | CIFAR-10 (Linear) | CIFAR-10 (k-NN) | CIFAR-100 (Linear) | CIFAR-100 (k-NN) |
> > |--|:-:|:-:|:-:|:-:|
> > |1.00|94.8|94.0|75.2|69.1|
> > |0.75|95.2|94.2|75.5|69.9|
> > |0.50|95.0|94.0|75.3|69.3|
> > |0.25|94.8|94.0|75.0|69.1|
> >
> > The best results are achieved at $\tau_s=0.1$, $\tau_t=0.04$, and $\alpha=0.75$ (as our adopted parameters in our paper) across both datasets and evaluation protocols. We will further add these results and include the corresponding brief explanations in the revised paper.
> >
> > [R1] CMID: A Unified Self-Supervised Learning Framework for Remote Sensing Image Understanding, TGRS, 2023.

---

> > > ### Comment · Reviewer_P5Pb · 2025-11-25
> > >
> > > My issue has been resolved — The Authors just need to add the revision details for the ImageNet results.
> > > That said, I'm still genuinely skeptical about achieving such strong performance with only one example, but out of appreciation for your effort, I'll raise the score to 6.

---

> ### Author Response · Authors · 2025-11-26
>
> Dear reviewer P5Pb,
>
> Thank you very much for your positive feedback and the updated score. We sincerely appreciate your constructive comments. We have updated the revised paper to include the revision details for the ImageNet results as requested. Thank you once again for your thoughtful review and valuable insights.
>
> Best regards,
>
> The Authors of Paper ID-6489

---

### Official Review · Reviewer_Jd4w · 2025-11-02

**Soundness:** 4
**Presentation:** 4
**Contribution:** 4
**Rating:** 6
**Confidence:** 3

**Summary:**

This paper introduces One-Shot Exemplar Self-Supervised Learning (OSESSL), a novel setting that leverages only one annotated instance per class to provide semantic grounding while maintaining the scalability of self-supervised learning. The authors propose a framework that constructs exemplar-guided prototypes augmented with discriminative neighbors from unlabeled data, and introduces exemplar-guided interpolation consistency to smooth decision boundaries. Extensive experiments on CIFAR and ImageNet benchmarks demonstrate state-of-the-art performance, with significant improvements in k-NN accuracy (e.g., +3% on CIFAR-100, +6% on ImageNet-100) over strong baselines.

**Strengths:**

**Originality**: The OSESSL setting is novel and fills an important gap between fully unsupervised SSL and semi-supervised learning. The exemplar-guided prototype construction and interpolation consistency mechanism are creative and well-motivated.

**Quality**: Experimental evaluation is extensive and convincing, covering linear evaluation, k-NN classification, semi-supervised learning, and transfer learning. The method consistently outperforms strong baselines across all settings.

**Clarity**: The paper is exceptionally clear in both writing and technical exposition. The gradient analysis provides valuable theoretical insight.

**Significance**: The work addresses the important problem of incorporating minimal supervision to guide SSL toward semantically meaningful representations, with practical implications for real-world applications where annotations are scarce.

**Weaknesses:**

1.	The method assumes clean exemplars are available, but real-world scenarios often involve noisy annotations. The paper briefly mentions this limitation but provides no experiments on noisy exemplars.

2.	The paper could benefit from more analysis on how the method performs with different qualities of exemplars (e.g., easy vs. hard examples).

**Questions:**

1.	How sensitive is the method to the quality of the single exemplar per class? Have you experimented with different strategies for selecting the exemplar (e.g., cluster centers vs. random samples)?

2.	For the neighbor selection in prototype construction, did you consider using more sophisticated metrics beyond cosine similarity, such as incorporating density estimation?

---

> ### Author Response · Authors · 2025-11-19
> **Response to Reviewer Jd4w (1/2)**
>
> Thank you for your positive and constructive comments! Our point-by-point responses are provided below.
>
> ---
>
> **Comment_1:** Analysis on noisy exemplars.
>
> **Response_1:** Thanks for your comment. While our setting follows the standard assumption of one clean exemplar per class, we agree with the reviewer that examining robustness under exemplar noise is also valuable. Therefore, we conduct additional experiments on CIFAR-10 and CIFAR-100 where we artificially corrupt exemplar labels with two types of noise: Permutation (a portion of exemplars swap labels among the class set) and Random (a portion of exemplars are assigned random labels from the full label space). We vary the noise ratio to 20%, 50%, and 80%. The results are reported below.
> | Type | Ratio | CIFAR-10 (Linear) | CIFAR-10 (k-NN) | CIFAR-100 (Linear) | CIFAR-100 (k-NN) |
> | --- | --- | :-: | :-: | :-: | :-: |
> | Permutation | 20% | 95.1 | 94.2 | 75.3 | 68.6 |
> | | 50% | 94.7 | 94.0 | 74.9 | 69.0 |
> | | 80% | 95.3 | 94.0 | 75.5 | 69.2 |
> | Random | 20% | 94.8 | 93.9 | 74.9 | 68.8 |
> | | 50% | 94.6 | 93.5 | 74.6 | 67.6 |
> | | 80% | 94.2 | 93.3 | 74.3 | 67.2 |
>
> As our method does not directly rely on exemplar labels for supervision, and those exemplars act as feature anchors, all prototypes are actually formed through neighbor-based aggregation. Accordingly, from the above results, we can easily observe that exemplar embeddings remain aligned with their semantic neighbors under permutation noise, leading to high robustness. In contrast, random noise breaks this alignment by assigning labels that are inconsistent with the exemplar's feature cluster, thus resulting in the corresponding performance drop. Meanwhile, we also compare our method with PAWS and Suave under the challenging random noise on CIFAR-10, and the results are presented below.
> | Method | Ratio | CIFAR-10 (Linear) | CIFAR-10 (k-NN) |
> | --- | --- | :-: | :-: |
> | PAWS | 20% | 92.8 | 89.5 |
> | | 50% | 92.1 | 88.1 |
> | | 80% | 90.2 | 87.1 |
> | Suave | 20% | 90.1 | 82.4 |
> | | 50% | 88.2 | 80.7 |
> | | 80% | 86.2 | 78.7 |
> | Ours | 20% | 94.8 | 93.9 |
> | | 50% | 94.6 | 93.5 |
> | | 80% | 94.2 | 93.3 |
>
> Under random noise corruption, both baselines exhibit substantially larger degradation than our method, where our method degrades much more mildly in all cases. This clearly demonstrates that our exemplar-guided prototype construction and discriminative weighting effectively dilute the impacts of noisy exemplars, thereby making the method inherently more robust.
>
> ---
>
> **Comment_2:** Sensitivity of exemplars.
>
> **Response_2:** Thanks for your comment. In fact, our method randomly selects exemplars with a simple default random seed, and here we want to follow the reviewer’s suggestion to conduct additional experiments to evaluate different selection strategies.
> We compare four exemplar-selection strategies: the samples closest to the cluster center (Easy_cluster), the samples with the highest confidence (Easy_conf), the samples farthest from the cluster center (Hard_cluster), and the samples with the lowest confidence (Hard_conf).
> | Type | CIFAR-10 (Linear) | CIFAR-10 (k-NN) | CIFAR-100 (Linear) | CIFAR-100 (k-NN) |
> | --- | :-: | :-: | :-: | :-: |
> | | Linear | k-NN | Linear | k-NN |
> |Easy_cluster|94.8|94.5|75.4|69.3|
> |Easy_conf|94.7|93.9|75.2|69.8|
> |Hard_cluster|94.8|93.9|75.3|69.0|
> |Hard_conf|94.8|94.1|75.5|69.4|
> |Ours|95.2|94.2|75.5|69.9|
>
> The experimental results show that different selection strategies finally yield similar performance, which implies that even hard or atypical exemplars do not really degrade the learned representations too much. This robustness arises because the exemplar is never used as a direct supervision target. Instead, its influence is mediated through neighbor aggregation and discriminative weighting, which together suppress the impact of suboptimal exemplars.

---

> > ### Author Response · Authors · 2025-11-19
> > **Response to Reviewer Jd4w (2/2)**
> >
> > **Comment_3:** Strategy of prototype construction.
> >
> > **Response_3:** Thanks for your comment. We follow the reviewer’s suggestion to incorporate local density into our neighbor selection process. Specifically, we compute a density estimate for each sample in the memory bank and then combine it with our discriminative score by multiplying the two terms. The corresponding recognition accuracy rates are listed below.
> > | Method | CIFAR-10 (Linear) | CIFAR-10 (k-NN) | CIFAR-100 (Linear) | CIFAR-100 (k-NN) |
> > | --- | :-: | :-: | :-: | :-: |
> > | Density | 94.8 | 93.9 | 74.9 | 68.8 |
> > | Ours | 95.2 | 94.2 | 75.5 | 69.9 |
> >
> > The density-based variant yields slightly lower performance compared to our method. This is because our discriminative score already captures the meaningful neighborhood structure which is ignored in density weighting. Actually, the simple density weighting tends to amplify high density regions without distinguishing semantically irrelevant neighbors from each other, yet our method can successfully separate them.
> >
> > An interesting future direction is to design more sophisticated neighbor-scoring mechanisms beyond cosine similarity or density scaling. These may include adaptive metrics based on exemplar semantics or jointly learned similarity functions with the representation. Such methods would enhance robustness by selectively emphasizing relevant neighbors or suppressing spurious matches.

---

### Comment · Area_Chair_Jjkd · 2025-11-19
**Start discussion**

Hi all,

The authors have submitted their response to the initial reviews, and we now enter the discussion phase.

Please review the authors' response and the comments from other reviewers. Based on the rebuttal and discussion, please update your final score if appropriate.

We welcome and encourage further discussion as needed.

Thank you for your continued contributions.

Best regards,

Your AC.

---

### Author Response · Authors · 2025-12-02
**Summary of The Initial Scores and Rebuttal**

Dear Area Chair,

Thank you for overseeing the review of our paper.

Our paper received initial scores of 8 (Reviewer Bjyr), 8 (Reviewer Ubdk), 6 (Reviewer Jd4w), 4 (Reviewer P5Pb), and 4 (Reviewer dQe4). The three positive reviewers reported higher confidence than the two negative reviewers. Overall, all reviewers acknowledged the motivation and results of our work, while two negative reviewers raised specific concerns.

To address these concerns, we have provided detailed clarifications and additional experiments in our rebuttal. Reviewer P5Pb and Bjyr explicitly indicated that their concerns were resolved, and Reviewer P5Pb updated his/her **score from 4 to 6** on Nov. 25 (UTC-3), prior to the information-leak incident on Nov. 27 (UTC-3).

Reviewer dQe4 did not have the opportunity to respond before the commenting window closed on Nov. 28 (UTC-3). The reviewer's concerns primarily focused on **the semi-supervised experimental details and the contribution of the OSESSL problem setting**. By carefully clarifying the core differences between the OSESSL task and the standard semi-supervised learning as well as providing all requested experimental details, we believe that our responses adequately address these points of Reviewer dQe4.

We would appreciate it if our clarifications and additional results could be taken into consideration during your final recommendation.

Thank you again for your time!

Best regards,

Authors of Paper ID-6489

---

### Meta-Review · Area_Chair_LSuS · 2026-01-06

**Summary:**

In this paper, the authors propose a new setting: One-Shot Exemplar Self-Supervised Learning, i.e., self-supervised learning that requires only one instance annotation per class. They further propose a framework that leverages the single-labeled exemplar to build the class-specific prototype for learning reliable representations from the huge unlabeled data, and build a novel consistency regularization. Experiments on CIFAR and ImageNet-100 are provided to show the effectiveness of the proposed method.

Most of the concerns were addressed during the rebuttal stage. But there are still some partially outstanding concerns that should be further explained. For example, Reviewer dQe4’s concern about the novelty of the OSESSL setting (vs. being a special case of semi-supervised learning). During the initial reviewing stage, this paper was reviewed by five expert reviewers, and three of them were positive. During the rebuttal period, Reviewer P5Pb upgraded their initial score from 4 to 6. While Reviewer dQe4 is the only reviewer who remains negative. I believe even if Reviewer dQe4's concerns are still partially outstanding, this paper is worthy of being presented at ICLR. The authors should carefully discuss the novelty of the OSESSL setting and explain the motivation for studying this setting in the final version.

**Reviewer Concerns:**

Most of the concerns were addressed during the rebuttal stage. But there are still some partially outstanding concerns that should be further explained. For example, Reviewer dQe4’s concern about the novelty of the OSESSL setting (vs. being a special case of semi-supervised learning).

**Reviewer Scores:**

During the initial reviewing stage, this paper was reviewed by five expert reviewers, and three of them were positive. During the rebuttal period, Reviewer P5Pb upgraded their initial score from 4 to 6. While Reviewer dQe4 is the only reviewer who remains negative. I believe even if Reviewer dQe4's concerns are still partially outstanding, this paper is worthy of being presented at ICLR. The authors should carefully discuss the novelty of the OSESSL setting and explain the motivation for studying this setting in the final version.

---

### Decision · Program_Chairs · 2026-01-26

Accept (Poster)